# NEUROMLR: Robust & Reliable Route Recommendation on Road Networks

**Jayant Jain**\*, **Vrittika Bagadia**\*, **Sahil Manchanda, Sayan Ranu**
Indian Institute of Technology Delhi
{jayantjain100,vrittikabagadia}@gmail.com
{sahil.manchanda,sayanranu}@cse.iitd.ac.in

## Abstract

Predicting the most likely route from a source location to a destination is a core functionality in mapping services. Although the problem has been studied in the literature, two key limitations remain to be addressed. First, our study reveals that a significant portion of the routes recommended by existing methods fail to reach the destination. Second, existing techniques are transductive in nature; hence, they fail to recommend routes if unseen roads are encountered at inference time. In this paper, we address these limitations through an *inductive* algorithm called NEUROMLR. NEUROMLR learns a *generative model* from historical trajectories by conditioning on three explanatory factors: the current location, the destination, and real-time traffic conditions. The conditional distributions are learned through a novel combination of *Lipschitz embeddings* with *Graph Convolutional Networks (GCN)* using historical trajectory data. Through in-depth experiments on real-world datasets, we establish that NEUROMLR imparts significant improvement in accuracy over the state of the art. More importantly, NEUROMLR generalizes dramatically better to unseen data and the recommended routes reach the destination with much higher likelihood than existing techniques.

## 1   Introduction and Related Work

Given historical trajectory data, we study the problem of predicting the *most likely route* from a source node to a destination node in a road network. This problem has two prominent applications: *route recommendation* and *route recovery*. *Route recommendation* is one of the core functionalities in GPS-aided mapping applications. They are routinely used in the cab and food-delivery industry [26, 7, 34, 6], as well as by common people through navigation systems when they are unfamiliar with their surroundings [30]. *Route recovery*, as the name suggests, focuses on recovering the actual traversed route from a partially observed GPS trajectory [9]. Due to various reasons, for instance, limiting the power consumption of GPS devices, trajectories are often recorded at low sampling rate.

### 1.1   Existing work

The simplest approach is to predict the shortest or the quickest path (route) between the source and destination. However, several studies have shown that human beings rarely travel in shortest paths [21]. Rather, the probability of a path being taken is a complex mixture of several *latent* factors such as road quality [5, 22], road scenery [21, 14], pollution levels [23], presence of road tolls, etc. Furthermore, it has also been shown that human beings systematically misperceive travel time[20]. Modeling these complex factors is challenging. Consequently, a large body of work exists on predicting the most likely route [1, 28, 33, 4, 10, 27], with DEEPST [10] and CSSRNN [27] being the best performing algorithms. CSSRNN models the trajectory patterns through a Recurrent Neural Network and exploits the topological constraints presented by the road network. DEEPST learns representations for the trajectories, destination and traffic conditions using variational autoencoders.

---

\*denotes equal contribution

35th Conference on Neural Information Processing Systems (NeurIPS 2021).

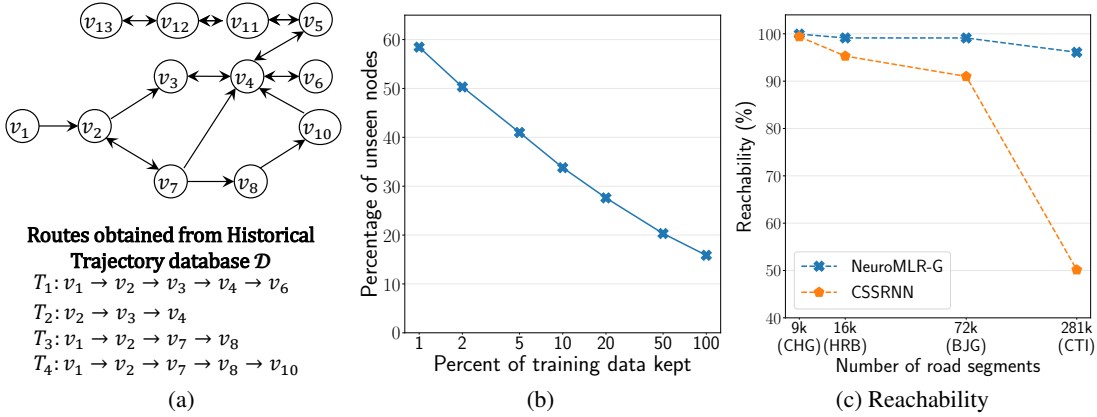

Figure 1: **(a)** A sample road network and historical database $\mathcal{D}$. For simplicity, we ignore edge weights. **(b)** Percentage of unseen nodes in the road network of Beijing against the size of the training data **(c)** Impact of road network size on performance of CSSRNN and NEUROMLR-G.

## 1.2  Limitations of existing work

• **Reachability:** Existing techniques have primarily used *recall* and *precision* to measure accuracy of predictions [10, 27]. Critically, these ignore whether the predicted trajectory actually reaches the destination. The deployability of any route recommendation algorithm is severely compromised if the recommended route does not reach the destination. Our experiments reveal that for both DEEPST and CSSRNN, a significant portion of recommended routes fail to reach the destination (See Table. 2). Further, the state of the art does not scale well to large road networks (see Fig. 1c). The performance gap between CSSRNN and NEUROMLR becomes more prominent with increasing network size.

• **Inductive Learning:** A good prediction model should be capable of making predictions on parts of the road network which were unseen or rarely seen during training. From Fig 1b we can observe that there is a significant percentage of nodes in the road network that remain unseen even if we use the entire training data. Both CSSRNN and DEEPST learn node embeddings in a transductive manner, limiting knowledge sharing across nodes. Hence, quality on unseen/lesser seen nodes suffers. DEEPST employs a clustering-inspired approach for sharing statistical strength across trips having similar destinations. However, this introduces a limitation that different destinations get mapped to identical representations, impairing reachability.

• **Prediction Accuracy:** The accuracy of even the best performing techniques is often below $50\%$ [10]. Consequently, there is scope for improvement.

## 1.3  Contributions

• **Decoupled route prediction:** Our proposed problem formulation (§ 2) allows decoupling the problem into two independent subproblems of route search (§ 3) and predicting transition probabilities (§ 4). This allows us to reduce the problem of finding the most likely route to that of identifying the shortest path in a modified road network, thus guaranteeing destination reachability (§ 3).

• **Generalization capability:** We propose an inductive learning method using a novel combination of *Lipschitz embeddings* with *Graph Convolutional Networks*. Lipschitz embeddings serve as a rich initialisation derived from the global road network structure, independent of historical data. GCNs, via message passing, further propagate information learnt during training, to unseen and rarely seen nodes.

• **Empirical Evaluation:** Extensive experiments on five large, real datasets establish: **(1)** NEUROMLR is up to $100\%$ more accurate and $4$ times faster, **(2)** the recommended route reaches destination with $> 0.96$ probability, which is up to $90\%$ better than the state of the art, and **(3)** NEUROMLR is dramatically more effective in generalizing to unseen data.

## 2  Problem Formulation

**Definition 1** (Road Network). *A road network is a directed graph $\mathcal{G} = (\mathcal{V}, \mathcal{E}, \delta, \tau_t)$, where $\mathcal{V}$ is the set of nodes representing road intersections, $\mathcal{E} \subseteq \mathcal{V} \times \mathcal{V}$ is the set of edges representing road*

*segments, a distance function $\delta : \mathcal{E} \rightarrow \mathbb{R}$ representing the length (weight) of each road segment, and function $\tau_t : \mathcal{E} \rightarrow \mathbb{R}$ representing the average time taken to traverse each road segment at time $t$.*

We use the notation $e = (u, v)$ to denote a road segment (edge) from node $u$ to $v$ and its length is denoted by $\delta(e)$.

**Definition 2** (Route). *A route (a.k.a path) $\mathcal{R}(s, d) = \{v_1, \cdots, v_k\}$ corresponds to a simple path from the source node $s = v_1$ to destination $d = v_k$ in the road network $\mathcal{G}$, i.e., path without cycles. Analogously, a route can also be expressed in terms of a sequence of edges $\mathcal{R}(s, d) = \{e_1, \cdots, e_{k-1}\}$, where $e_i = (v_i, v_{i+1})$.*

To denote a generic route between any arbitrary source and destination nodes, we use $\mathcal{R}$ instead of $\mathcal{R}(s, d)$. We use the notation $|\mathcal{R}|$ to denote the number of edges in $\mathcal{R}$ and $R.e_i$ to refer to the $i^{th}$ edge in $\mathcal{R}$. Furthermore, $e \in \mathcal{R}$ denotes that $\mathcal{R}$ goes through edge $e$. The above notations are analogously extended from edges to nodes.

We assume we have access to a dataset of *historical trajectories*. A trajectory is a sequence of GPS pings, made by a vehichle, which corresponds to a path in the road network. Each GPS ping is a tuple of the form $\langle latitude, longitude, time \rangle$, which records a vehicle's location at a particular time.

**Definition 3** (Query). *In a route recommendation query, the input is a tuple $q : \langle s, d, t \rangle$, where $s, d \in \mathcal{V}$ are the source and destination nodes respectively, and $t$ is the time at which the journey is to be taken.*

**Problem: Most Likely Route:** *Given a road network $\mathcal{G}$, a historical database $\mathcal{D}$ of trajectories, and query $q : \langle s, d, t \rangle$, we would like to infer the mostly likely route $\mathcal{R}^*(s, d)$ based on the traffic patterns embodied in $\mathcal{D}$. Formally,*

$$\mathcal{R}^*(s, d) = \arg \max_{\forall \mathcal{R}(s,d) \in \mathcal{G}} Pr(\mathcal{R}(s, d) \mid q) \tag{1}$$

## 2.1 Problem Characterization

The route recommendation problem can be framed as a path search problem on the graph corresponding to the road network. Mathematically, the probability of a route $\mathcal{R}$ can be expressed in terms of its constituent edges.

$$Pr(\mathcal{R} \mid q) = \prod_{i=1}^{|\mathcal{R}|} Pr(\mathcal{R}.e_i \mid \mathcal{R}.e_0 \rightarrow \mathcal{R}.e_{i-1}, s, d, t) \tag{2}$$

Here, $Pr(\mathcal{R}.e_i \mid \mathcal{R}.e_0 \rightarrow \mathcal{R}.e_{i-1}, s, d, t)$ represents the probability that route $\mathcal{R}$ goes through edge $\mathcal{R}.e_i$ given the path taken so far and the query parameters $q : \langle s, d, t \rangle$ Past studies have shown that human mobility patterns conform to the *Markovian* assumption [1, 25]. Thus, Eq. 2 reduces to:

$$Pr(\mathcal{R} \mid q) = \prod_{i=1}^{|\mathcal{R}|} Pr(\mathcal{R}.e_i \mid v_i, d, t) \tag{3}$$

With these simplifications, Eq. 1 reduces to:

$$\mathcal{R}^*(s, d) = \arg \max_{\forall \mathcal{R} \in \mathcal{G}} \prod_{i=1}^{|\mathcal{R}|} Pr(\mathcal{R}.e_i \mid v_i, d, t) \quad = \arg \min_{\forall \mathcal{R} \in \mathcal{G}} \sum_{i=1}^{|\mathcal{R}|} -\log(Pr(\mathcal{R}.e_i \mid v_i, d, t)) \tag{4}$$

To summarize the above observations, there are two key challenges that we need to tackle:

• **Route Search:** Given the conditional transition probability distribution, searching the route (path) minimizing Eq. 4.

• **Modeling:** Learning the conditional distribution that governs transitions in the road network, from the historical trajectory dataset $\mathcal{D}$.

Henceforth, while talking about transitions in the road network, we will use the notation $curr$ to refer to the current location(node), $curr \in \mathcal{V}$.

## 3 Route Search

In this section, we assume that the transition probability $Pr(e|curr, d, t)$ for any edge $e \in \mathcal{E}$ is known. Our algorithm to learn this distribution is discussed in § 4.

**The Optimal Search:** We first note that the negative log likelihood of a transition probability (see Eq. 4) would be non-negative. Thus, we have an edge weight for each edge in the road network and

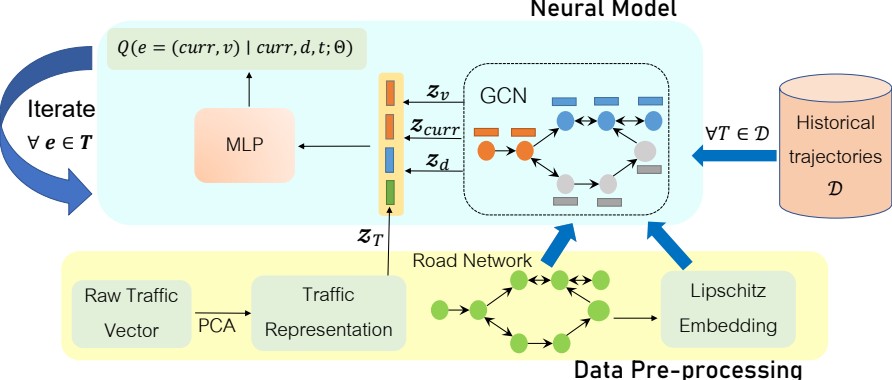

Figure 2: Architecture of NEUROMLR.

our goal is to identify the path from the source to the destination that has the minimum cumulative weight. This computational task maps to the problem of finding the shortest path in a graph and can be solved using *Dijkstra's* Algorithm. More importantly, the optimal path is *guaranteed* to reach the destination. The pseudocode for this search algorithm is provided in App. A.

*Computational Complexity:* The complexity of the Dijkstra's Algorithm is $O(|\mathcal{V}| + |\mathcal{E}|\log(|\mathcal{V}|))$. This may be prohibitively expensive on large road networks where real-time predictions are desired.

**Greedy Approach:** We start from the source node and greedily choose the transition with the highest probability till the destination is reached. Inaccurate estimation of transition probabilities may however divert us towards the wrong direction and we may never reach the destination. This may result in $|\mathcal{V}|$ iterations in the worst case. To handle cases of this nature, we terminate when either the destination is reached or the Haversine distance from the current node to the destination is significantly higher than the closest point in the current route to the destination. This idea is motivated from the fact that, in general, a vehicle progressively moves closer to the destination with each transition [31]. The pseudocode of the greedy approach can be found in Alg. 2 in App. B .

*Computational Complexity:* At each node in $\mathcal{R}^*$, we evaluate each neighbor and select the one with highest likelihood. Hence, the complexity is $O(g|\mathcal{R}^*|)$, where $g$ is the average degree in $\mathcal{G}$.

## 4 NEUROMLR: The Neural Approach to the Most Likely Route Problem

Revisiting Eq. 4, the key requirement is to accurately model the conditional transition probability function $Pr(e \mid curr, d, t)$ where $e = (curr, v)$. We want to estimate the *true* distribution that governs transitions in the road network. However, this distribution is hidden from us and we only have access to $\mathcal{D}$, which is a sample drawn from this distribution.

Mathematically, we wish to estimate the underlying transition probability distribution from $\mathcal{D}$ using a surrogate function $Q(n|c, d, t; \Theta)$, such that, $Q(e|curr, d, t; \Theta) \approx Pr(e|curr, d, t)$.

We learn $\Theta$ using the neural network depicted in Fig. 2. Our core idea is to learn useful representations of road intersections (nodes) and real-time traffic conditions, and use them to infer the transition probabilities. To learn inductive node representations, we use a novel combination of GCN [8] with *Lipschitz embeddings* [2]. In addition, a low-dimensional traffic representation of the road network at any time $t$ is learned using Principal Component Analysis (PCA). To predict $P(e = (curr, v) \mid curr, d, t)$, we concatenate the representations of $v$, $curr$, $d$ and traffic status at time $t$, and pass them through a *Multi-layered Perceptron (MLP)* to predict the transition probability. The entire network is trained end-to-end. We next discuss each of the sub-components. By convention, we used bold font for vectors and matrices.

### 4.1 Constructing Node Attributes

In this section, we describe the process of constructing node attributes for our GCN. We wish to learn embeddings where nodes with similar routes to common destinations are close in the embedding space. While latitude and longitude may be used as node attributes, they do not characterize node positions accurately since movement of vehicles is constrained by the network structure. Rather, we need to learn node attributes that reflect road network distances. Towards that end, we use *Lipschitz embeddings*.

**Definition 4** (Attribute Embedding). *Let $\mathcal{A} = \{a_1, \cdots, a_k\} \subseteq \mathcal{V}$ be a randomly selected subset of nodes. We call them anchors. The distance $d(u, v)$ between two nodes $u, v \in \mathcal{V}$ is defined as $\frac{sp(u,v)+sp(v,u)}{2}$, where $sp(u, v)$ is the spatial shortest path distance[2] from $u$ to $v$. We embed all nodes in $\mathcal{V}$ in a $k$-dimensional feature space $\boldsymbol{\nu_L}(u) = [x_1, \cdots, x_k]$, where $x_i = d(u, a_i)$.*

The dimensionality of the attribute space dictates how well the original shortest path distances are preserved. To gain a formal understanding of distance preservation, we introduce the definition of *distortion*.

**Definition 5** (Distortion). *Given two metric spaces $(\mathcal{O}, d)$ and $(\mathcal{O}', d')$ and an embedding function $f : \mathcal{O} \to \mathcal{O}'$, $f$ has a distortion $\alpha$ if $\forall o_1, o_2 \in O$, $\frac{1}{\alpha} d(o_1, o_2) \le d'(f(o_1), f(o_2)) \le d(o_1, o_2)$.*

In our case, $d(o_1, o_2)$ is the average two-way shortest path distance. $d'()$, along with dimensionality $k$, remains to be defined. To define them, we use *Bourgain's Theorem*. Bourgain's Theorem [2] establishes that a low *distortion* Lipschitz embedding exists for any metric space.

**Theorem 1** (Bourgain's Theorem [2]). *Given any finite metric space $(\mathcal{O}, d)$ with distance function $d(\cdot)$, there exists an embedding of in $(\mathcal{O}, d)$ into $\mathbb{R}^k$ under any $l_p$ metric, where $k = O(\log^2 n)$ and the distortion of the embedding is $O(\log n)$, where $n = |O|$.*

To apply Bourgain's Theorem on our problem, we need to show that $d(u, v)$ is metric.

**Lemma 1.** *$d(u, v)$ is a metric distance function. For proof, refer App. C*

## 4.2 Learning Node Representations through GCN

In addition to capturing road network distances in node representations through *Lipschitz embeddings*, we would also like to *generalize to unseen nodes* in the road network. To illustrate, let us assume node $v_{13}$ in Fig. 1a has not appeared in any training trajectory. If we fine-tune representations for only seen nodes, then the representation of $v_{13}$ would remain unchanged from its Lipschitz embedding. A GCN avoids this scenario by *message passing* among neighbors. More specifically, information is shared among $L$-hop neighbors in GCN, where $L$ is the number of layers. Thus, if a subset of these neighbors have appeared in training trajectories, then this information is shared in its neighborhood, which infuses information beyond Lipschitz embeddings even for unseen nodes.

To train the GCN, Lipschitz embeddings $\boldsymbol{\nu}(\cdot)$ corresponding to $|\mathcal{V}|$ nodes are stacked as original input features. Specifically, $\forall u \in \mathcal{V}$, $\boldsymbol{h_u^0} = \boldsymbol{\nu}(u)$. Next, we compute:

$$\boldsymbol{h_u^l} = \sigma \left( \boldsymbol{W_l} \sum_{v \in N(u) \cup u} \frac{\boldsymbol{h_v^{l-1}}}{\sqrt{(|N(v)| + 1)(|N(u)| + 1)}} \right) \tag{5}$$

Here, $\boldsymbol{W_l}$ stands for layer-specific learnable weight matrix for $l^{th}$ layer, $\boldsymbol{h_u^l}$ is the embedding of node $u$ at layer $l$ and $\sigma(\cdot)$ denotes an *activation function* (ReLU in our implementation). Furthermore, $N(u) = \{v \in \mathcal{V} \mid (u, v) \in \mathcal{E}\}$ denotes the neighbors of node $u$ in the road network. The vector formed in the final layer $L$ is $\boldsymbol{z_u} = \boldsymbol{h_u^L}$.

## 4.3 Traffic representation

The simplest option is to partition $\mathcal{D}$ into various time slots (Ex: 8AM-11AM, 11-AM-2PM, etc.) and learn a model separately for each time slot. This strategy, however, assumes that traffic is homogeneous in each time slot on all days. Clearly, this assumption is not true in real life (Ex: weekends vs. week days). Further, this scheme is data inefficient since the traffic-dependent travel patterns across different time-slots might be similar.

In order to characterize the real-time traffic conditions for the entire road network at time $t$, we obtain the top-5% of the most frequently traversed edges $E \subseteq \mathcal{E}$ in $\mathcal{D}$. The *raw* traffic representation at time $t$ is the $|E|$-dimensional vector of speeds on these edges. Specifically, $\boldsymbol{r_t} = [\tau_t(e) \mid e \in E]$. We use only the top-5% edges since the frequency distribution of edges follows a power-law (See App. H). Consequently, we do not use noisy speed information on less-travelled edges.

It is natural for neighboring road segments (edges) to exhibit co-variance in speed. To remove such information redundancy, we learn a low-dimensional representation of $\boldsymbol{r_t}$ through PCA. Specifically, let the training trajectories in $\mathcal{D}$ span the time range $[t_{min}, t_{max}]$. Thus, we have a collection of

---

[2]We assume that the road network is strongly connected, which is typically true.

traffic representations in the form of $R = \{r_t \mid t \in [t_{min}, t_{max}]\}$. We perform PCA to learn the eigenvectors on $R$. Given any raw traffic vector $r_t$, it is projected on the top-$k$ eigenvectors to construct the low-dimensional representation $z_t$. Mathematically, $z_t = r_t V$, where $V \in \mathbb{R}^{|E| \times k}$. The columns of $V$ contain the eigenvectors with the $k$ largest eigenvalues.

### 4.4 Aggregating Node embedding with Traffic representation

$Q(e|curr, d, t; \Theta)$ is a function of four input features: current node $curr$, destination $d$, traffic condition at time $t$, and the transition node $v$ corresponding to edge $e = (curr, v)$. GCN provides the embeddings $z_{curr}$, $z_v$, $z_d$ of $curr$, $v$, $d$ respectively and the traffic representation $z_t$ is constructed as discussed above. All these embeddings are concatenated [3] as :

$$z = [z_{curr}||z_v||z_d||z_t] \tag{6}$$

### 4.5 Model Training

Following the pass through GCN, the concatenated embedding $z$ (Eq. 6) is passed through an MLP to convert the vector into a scalar (unnormalised) *confidence* value i.e $f(v, curr, d, t) = MLP(z)$. The predicted transition value is defined as a softmax over all possible transitions from $curr$. Mathematically,

$$Q\left((curr, v)|curr, d, t; \Theta\right) = \frac{exp\left(f\left(v, curr, d, t\right)\right)}{\sum_{v' \in N(curr)} exp\left(f\left(v', curr, d, t\right)\right)} \tag{7}$$

$$Loss(\Theta) = -\frac{1}{|\mathcal{D}|} \sum_{\forall \mathcal{R} \in \mathcal{D}} \sum_{j=1}^{|\mathcal{R}|} log\, Q(\mathcal{R}.e_j | \mathcal{R}.v_j, \mathcal{R}.d, \mathcal{R}.t; \Theta) \tag{8}$$

Finally, in Eq. 8 above, the model parameters $\Theta$ are optimized through *cross-entropy* loss over trajectories in $\mathcal{D}$. The pseudocode of the training procedure can be found in Alg. 3 in App. D.

### 4.6 Inference Phase

Given any query $q : \langle s, d, t \rangle$, we follow one of the route search algorithms discussed in § 3. Both search mechanisms require only the transition probabilities as edge weights. Prediction of a transition probability $Pr(e = (curr, v) \mid curr, d, t)$ simply involves a forward pass through the architecture.

**Complexity analysis:** Time complexity of predicting $Pr(e = (curr, v) \mid curr, d, t)$ is $O(g \cdot L_M \cdot d_f^2)$. Here, $L_M$ is the number of the layers in the MLP, $g$ is the average degree in the road network, $d_f$ is the output feature vector dimension of the GCN. Refer Appendix. E for detailed derivation.

## 5 Experiments

In this section, we benchmark NEUROMLR against DEEPST and CSSRNN and establish that:

- **Accuracy:** NEUROMLR is more accurate in terms of precision and recall when compared to the state-of-the-art algorithms of DEEPST [10] and CSSRNN [27].
- **Reachability:** NEUROMLR, with its greedy route search mechanism, is more efficient, and achieves significantly higher reachability than DEEPST and CSSRNN.
- **Inductive Learning:** Due to its inductive learning capability, NEUROMLR learns more effectively and generalizes significantly better to unseen/lesser seen parts of the road network.
- **Scalability:** NEUROMLR generates high quality routes on large road networks. The performance of CSSRNN, on the contrary, deteriorates heavily with increase in road network size(Fig. 1c).

Our code-base is available at `https://github.com/idea-iitd/NeuroMLR`.

### 5.1 Experimental Setup

The system configuration details are present in App. F. All experiments are repeated 5 times and we report the average of the metric being measured.

**Datasets**: We use publicly available real datasets from five different cities. Table 1 summarizes the statistics of the datasets. The first four cities namely Chengdu[4], Porto[15], Harbin [11] and

---

[3]We also tried the Attention mechanism [24] to capture the importance of each of the explanatory factors. Details can be found in Appendix. N.

[4]Chengdu dataset link

| Statistics | Chengdu(CHG) | Porto(PT) | Harbin(HRB) | Beijing(BJG) | CityIndia(CTI) |
|---|---|---|---|---|---|
| No. of nodes | $3,973$ | $5,330$ | $6,598$ | $31,199$ | $105,873$ |
| No. of edges | $9,255$ | $11,491$ | $16,292$ | $72,156$ | $281,086$ |
| No. of trajectories | $3,600,503$ | $1,426,312$ | $1,133,548$ | $1,382,948$ | $451,443$ |
| Avg trip length (km) | $4.54$ | $5.27$ | $10.92$ | $7.39$ | $3.27$ |
| Avg number of edges/trip | $22.93$ | $51.07$ | $56.81$ | $36.08$ | $42.68$ |

Table 1: Dataset statistics after pre-processing.

| Algorithm | Precision (%) | | | Recall (%) | | | Reachability (%) | | | Reachability Distance(km) | | |
|---|---|---|---|---|---|---|---|---|---|---|---|---|
| | HRB | BJG | CTI | HRB | BJG | CTI | HRB | BJG | CTI | HRB | BJG | CTI |
| NEUROMLR-D | **66.1** | **77.9** | **77.9** | 49.6 | **76.5** | **73.1** | — | — | — | — | — | — |
| NEUROMLR-G | 59.6 | 75.6 | 74.3 | 48.6 | 74.5 | 70.1 | **99.1** | **99.1** | **96.1** | **0.02** | **0.01** | **0.03** |
| CSSRNN | 49.8 | 59.5 | 36.9 | **51.1** | 68.8 | 53.2 | 95.3 | 91.7 | 50.2 | 0.16 | 0.83 | 2.03 |
| DEEPST | 51.9 | 60.3 | 67.4 | 27.3 | 33.2 | 34.9 | 8.1 | 8.7 | 6.7 | 1.96 | 2.75 | 1.07 |
| SP | 46.4 | 59.2 | 62.1 | 31.3 | 55.5 | 53.7 | — | — | — | — | — | — |
| QP | 40.7 | 51.4 | 47.6 | 28.6 | 50.0 | 44.0 | — | — | — | — | — | — |

Table 2: Comparison of NEUROMLR against the benchmarked algorithms on the four different metrics of average precision, average recall, percentage of trips that reached destination, and average distance from the true destination. The best performance for each dataset is highlighted in bold.

Beijing[12] are taxi datasets. The fifth dataset is a publicly available food delivery dataset[7]. The authors did not reveal the name of the city, other than the fact that, in terms of food delivery volume, this city ranks among the highest in India. We extract the road network of each city from OpenStreetMap [17]. To align the GPS sequences to the road network we use map-matching[29]. The temporal edge weight $\tau_t(e)$ is set to the average travel time of all vehicles going over $e$ in the past one hour.

**Baselines:** We benchmark the performance of NEUROMLR against, **(1)** DEEPST, **(2)** CSSRNN, **(3)** *shortest path (SP)*, and **(4)** the *quickest path (QP)*. For NEUROMLR, we consider two versions: NEUROMLR-Dijkstra **(NEUROMLR-D)** and NEUROMLR-Greedy **(NEUROMLR-G)** corresponding to the two route search algorithms described in § 3. We do not consider personalized route recommendation algorithms [25], since we do not consider the personalization aspect. The codebase of CSSRNN, shared by the authors, is implemented in TensorFlow 1.15. DEEPST and NEUROMLR are implemented in PyTorch 1.6.0.

**Train-Validation-Test setup:** For a fair comparison of NEUROMLR with DEEPST and CSSRNN, we train all models till convergence of the validation loss. Before splitting, we sort the trajectories on the basis of the start timestamp. Unless specifically mentioned, we use the first $60\%$ of the trajectories for training, next $20\%$ for validation and remaining $20\%$ for inference.

**Evaluation metrics:** For evaluation, we pick each route $\mathcal{R}$ in the test set and issue the corresponding query $q : \langle s, d, t \rangle$, where $s = \mathcal{R}.v_1$, $d = \mathcal{R}.v_{|\mathcal{R}|+1}$, and $t$ is the time at which $\mathcal{R}$ was initiated. The predicted trajectory $\mathcal{R}^*$ for $q$ is then generated and compared with $\mathcal{R}$. We use four metrics to evaluate prediction quality.

$$\text{Precision} = \frac{\sum_{e \in (\mathcal{R} \cap \mathcal{R}^*)} \delta(e)}{\sum_{e \in \mathcal{R}^*} \delta(e)}, \qquad \text{Recall} = \frac{\sum_{e \in (\mathcal{R} \cap \mathcal{R}^*)} \delta(e)}{\sum_{e \in \mathcal{R}} \delta(e)}$$

$$\text{Reachability} = \begin{cases} 1 & \text{if } d = d^* \\ 0 & otherwise \end{cases}, \qquad \text{Reachability Distance} = Haversine\,(d, d^*)$$

In the above equations, $\delta(e)$ denotes the length of edge $e$ and $d^*$ denotes the last node in $\mathcal{R}^*$. Note that the reachability metrics are relevant for only NEUROMLR-Greedy, DEEPST and CSSRNN since the rest of the algorithms guarantee reachability.

**Parameters:** The default parameters for NEUROMLR are provided in App. G.

## 5.2 Accuracy and Reachability

Table 2 presents the performance of the various algorithms across the three larger datasets - Harbin, Beijing and City India[5]. The following observations emerge from Table 2.

---

[5]Due to space limitations, the results for the other datasets (Porto, Chengdu) can be found in App. L

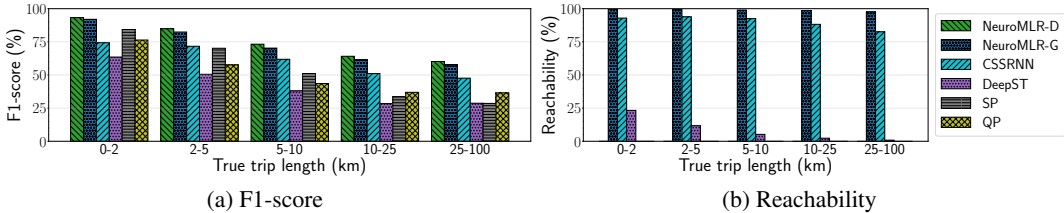

(a) F1-score                                    (b) Reachability

Figure 3: Impact of trip length on accuracy and reachability on the Beijing dataset.

**Precision and Recall:** Both versions of NEUROMLR consistently outperform the baselines[6]. As expected, NEUROMLR-Dijkstra is marginally better than NEUROMLR-Greedy. Among the four considered baselines, CSSRNN achieves the best result. Furthermore, NEUROMLR significantly outperform both SP and QP, which validates past work that people rarely follow shortest/quickest paths: automobile travellers select routes based on several latent factors such as the presence of tolls [16, 19], travel time reliability [3], traffic lights [18], and road aesthetics [21, 32].

**Reachability:** NEUROMLR-Greedy outperforms both DEEPST and CSSRNN and achieves more than 96% reachability across all datasets. This superior performance of NEUROMLR-Greedy establishes the efficacy of Lipschitz embeddings with GCN in modeling transitions that govern vehicle movements. The reachability performance is weakest in DEEPST due to the clustering based approach it adopts. Specifically, several nodes are allotted the same destination representation and hence reachability is compromised.

**Impact of Trip Length:** To further evaluate robustness, we examine the performance against the distance between the source and destination in the query. As the trip length increases, more possible paths emerge, and the modeling task gets harder. This pattern is visible in Fig. 3 [7]. Consistent with previous results, NeuroMLR continues to be the most accurate algorithm.

## 5.3   Inductive Learning

To showcase the benefits of inductive learning, we compare the two best performing algorithms of NEUROMLR with CSSRNN[8].

**Impact of training data:** In Figs. 4a- 4b, we examine how effectively each algorithm learns as the volume of training data is varied. As clearly evident, NEUROMLR obtains significantly more accurate performance at low volumes of training data than CSSRNN. This is a direct consequence of the inductive ability to share information among nodes and thereby generalize for unseen test data.

**Performance on unseen data:** In Figs. 4c-4i [9], we investigate how NEUROMLR and CSSRNN perform on test trips that originate or end at *unpopular* nodes. A node is termed *unpopular* if its frequency of occurrence in the training trips is less than a threshold frequency. We segregate the test trips into four categories based on the popularity of source and destination nodes: **P-P**, **U-P**, **P-U** and **U-U**. For example, **P-U** refers to trips starting at a *popular* node and ending at an *unpopular* one. As expected, the majority of test trips are of type **P-P** and thus the performance on **P-P** (Figures 4d, 4g) is similar to the aggregate results mentioned in Table 2. We have not included the performance variation of trips of type **U-U** for different thresholds, since the percentage of such trips is relatively insignificant (Fig.4c). The performance on **P-U** trips (Figs. 4i, 4f) undergoes a dramatic drop for CSSRNN since the destination is *unpopular* and there is not much information to direct the model. For **U-P** (Figs. 4h, 4e), the performance lies between **P-P** and **P-U** since even after a rocky start, it could transit to a *popular* node and from there may reach the destination. In all cases NEUROMLR adapts more gracefully and highlights the benefits of inductive learning. Note that at zero threshold frequency, unpopular nodes are equivalent to nodes unseen during training.

---

[6]In absolute terms, the precision and recall might seem low. This is because, for the same input (source-destination pair), we may have different outputs (routes). For more details, refer to App. P.

[7]For similar analysis on other datasets, refer to App. I

[8]Here, we have restricted our model's comparisons to CSSRNN since DEEPST performed poorly on all metrics. Some potential reasons can be found in App. M.

[9]For similar analysis on other datasets, refer to App. O.

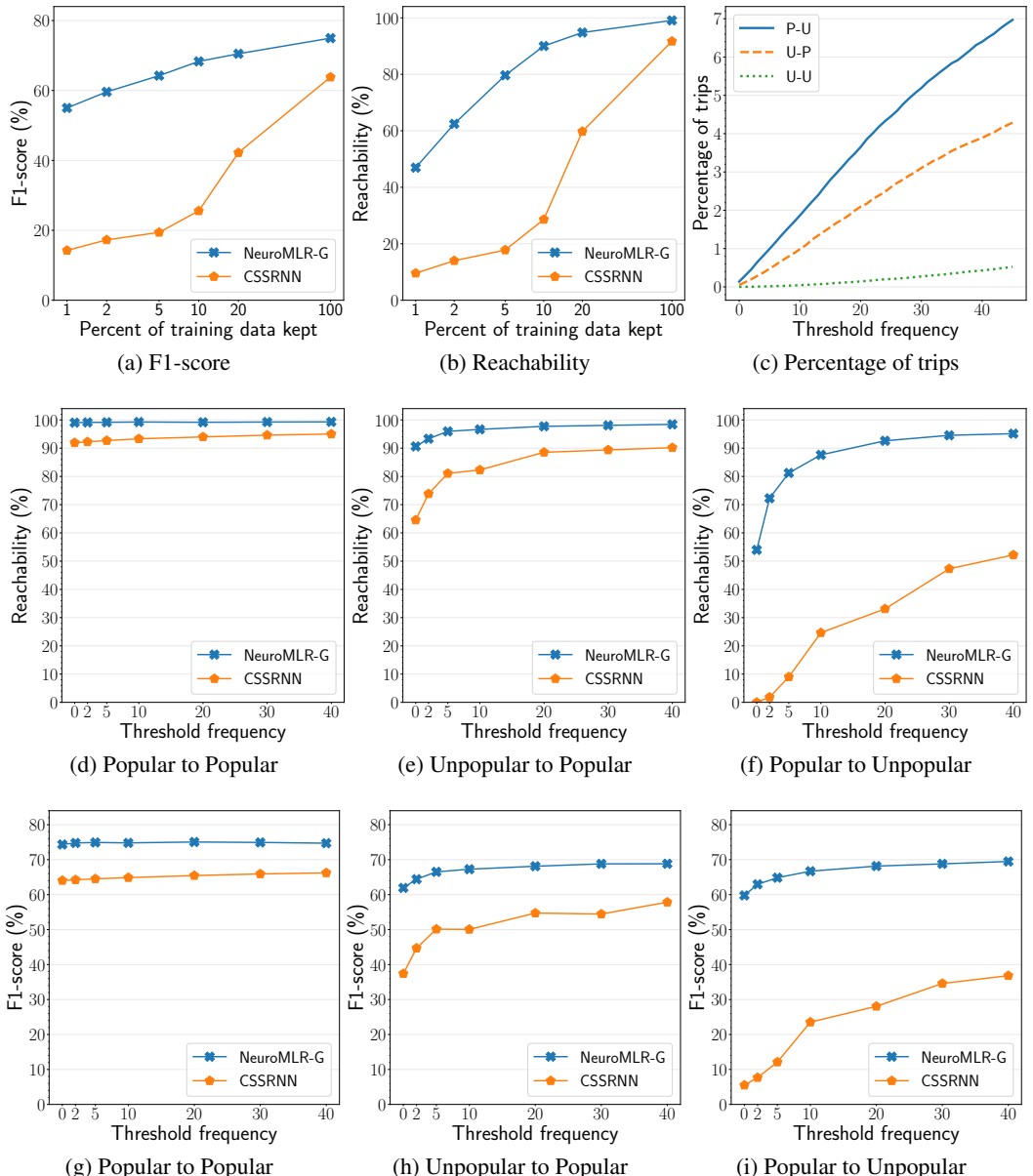

Figure 4: All these experiments are performed on the Beijing dataset. **(a, b)** Performance of NEUROMLR-G and CSSRNN with different percentages of training data. **(c)** Percentage of test data trips in **P-U**, **U-P** and **U-U** categories. **(d-i)** Variation of Reachability & F1-score with the threshold frequency for different popularity categories.

## 5.4 Ablation study

**Impact of GCN and Lipschitz Embeddings:** We investigate the individual impact of using Lipschitz embeddings and employing GCNs in Figs. 5a-5b by comparing the performance of NEUROMLR on the four possible combinations. As visible, the combined combination of Lipschitz embeddings and GCN imparts a significant improvement in both prediction accuracy and reachability. Employing either one individually also enhances the model's performance, thereby justifying their importance. For similar studies on other datasets see App. J.

**Impact of Traffic:** The impact of traffic on route prediction can be found in App. J.

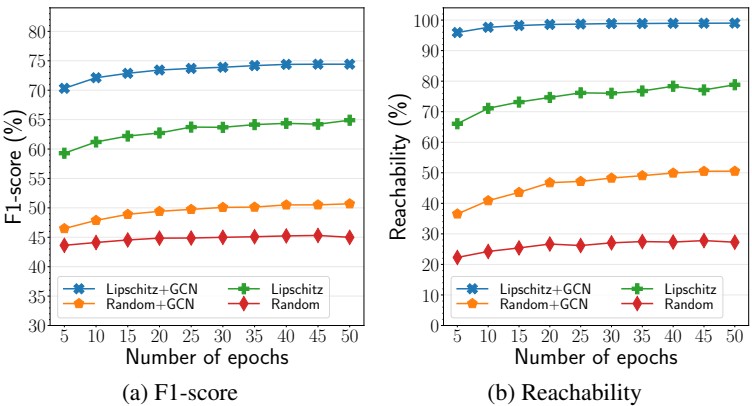

|                   | (a) F1-score | (b) Reachability |
|-------------------|:------------:|:----------------:|

Figure 5: Impact of GCN on further fine-tuning Lipschitz embeddings in NEUROMLR-Greedy. All plots in this figure use the Beijing dataset.

## 5.5 Inference Time

Table 3 presents the number of queries solved by the three best performing algorithms per second. Since the performance of DEEPST was found to be much inferior to these three algorithms (Table 2), we skip its efficiency analysis. As evident, NEUROMLR-Greedy is the fastest. Consistent with its computational complexity, NEUROMLR-Dijkstra is slower. CSSRNN is slower than NEUROMLR-Greedy since it uses a more complex auto-regressive model where each transition is also conditioned on the path taken so far. Owing to significantly faster inference and accuracy that is almost identical to NEUROMLR-Dijkstra, our recommendation is to use NEUROMLR-Greedy as the default algorithm.

| Algorithm  | Chengdu | Porto | Harbin | Beijing | CityIndia |
|------------|---------|-------|--------|---------|-----------|
| NEUROMLR-D | 11.38   | 10.11 | 7.62   | 1.08    | 0.28      |
| NEUROMLR-G | **5612** | **4539** | **3286** | **2084** | **1522** |
| CSSRNN     | 4425    | 1684  | 1355   | 522     | 467       |

Table 3: Inference speed (trips processed per second).

## 6 Conclusion

For a route recommendation algorithm to be deployable in the real world, it must ensure that the recommended route reaches the destination. In addition, it must show good generalization performance on queries over unseen/rarely seen data. Existing techniques for predicting the most likely route lack the above mentioned abilities. In this paper, we propose NEUROMLR which overcomes these limitations through a novel combination of *Lipschitz embedding* and *Graph Convolutional networks*. This strategy ensures inductive learning and enhances reachability. Specifically, even those nodes that are not seen adequately in training data, get good representations due to Lipschitz embedding capturing network position and GCN ensuring information propagation from neighboring nodes. Further, the model decouples probability transition modelling and route search, allowing us to use popular route search algorithms to guarantee reachability. All-in-all, NEUROMLR is more reliable, scales to larger cities, robust to unseen data and more effective in learning from low volume of data.

**Limitations:** In the future, we would like to work on capturing the personalization aspect of the problem and learn to transfer knowledge from one city to another.

**Potential for Negative Societal Impact:** Our proposed work facilitates robust and reliable computation of the mostly likely route in road networks. To the best of our understanding, we do not see any potential of negative societal impact from this work.

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
