# Appendix

## A  Optimal Path Search

For the optimal route search we want to find the shortest path in a modified graph. We adapt the standard Dijkstra's algorithm for the same. A priority queue is maintained for all the unvisited nodes. Each unvisited node is mapped to the minimum cost to reach that node from the source node, using the visited ones. In each iteration the minimum element is popped from the queue and it's neighbours are updated using the negative log likelihoods found from the model and added to the queue (lines 7-11).

Note that computing all the edge weights apriori instead of when needed might turn out to be faster, since a GPU might be able to compute all weights in one forward pass faster than individual forward passes in each iteration.

---

**Algorithm 1:** Optimal Route Search

**Input**    : Query $q : \langle s, d, t \rangle$
**Output**  : Most likely path $\mathcal{R}^*$

1  **for** $each\ v \in \mathcal{V}$ **do**
2  $\quad$ $dist[v] \leftarrow \infty$
3  $\quad$ $parent[v] \leftarrow None$
4  $dist[s] \leftarrow 0$
5  $Q \leftarrow \mathcal{V}$
6  **while** $Q \neq \emptyset$ **do**
7  $\quad$ $u \leftarrow \text{EXTRACT-MIN}(Q)$
8  $\quad$ **for** $each\ (u, v) \in \mathcal{E}$ **do**
9  $\quad\quad$ **if** $dist[v] > dist[u] - \log\left(Pr((v, u) \mid u, d, t)\right)$ **then**
10 $\quad\quad\quad$ $dist[v] = dist[u] - \log\left(Pr((v, u) \mid u, d, t)\right)$
11 $\quad\quad\quad$ $parent[v] \leftarrow u$
12 $\mathcal{R}^* \leftarrow \{d\}$
13 $curr \leftarrow d$
14 **while** $curr \neq s$ **do**
15 $\quad$ $curr \leftarrow parent[curr]$
16 $\quad$ $\mathcal{R}^* \leftarrow \{curr\} \cup \mathcal{R}^*$
17 **return** $\mathcal{R}^*$

---

## B  Greedy Route Search

---

**Algorithm 2:** Greedy Route Search

**Input**    : Query $q : \langle s, d, t \rangle$, hyper-parameter $\Delta$
**Output**  : Most likely path $\mathcal{R}^*$

1  $\mathcal{R}^* \leftarrow \emptyset$ , $curr \leftarrow s$
2  $currDist \leftarrow Haversine(curr, d)$
3  $minDist \leftarrow currDist$
4  **while** $curr \neq d$ **and** $currDist \leq \Delta \times minDist$ **do**
5  $\quad$ $\mathbb{N} \leftarrow \{v \in \mathcal{V} \mid v \notin \mathcal{R}^*,\ (curr, v) \in \mathcal{E}\}$
6  $\quad$ $curr \leftarrow \arg\max_{v \in \mathbb{N}} \{Pr((curr, v) \mid curr, d, t)\}$
7  $\quad$ $\mathcal{R}^* \leftarrow \mathcal{R}^* \cup \{v\}$
8  $\quad$ $currDist \leftarrow Haversine(curr, d)$
9  $\quad$ **if** $minDist < currDist$ **then**
10 $\quad\quad$ $minDist \leftarrow currDist$
11 **return** $\mathcal{R}^*$

---

## C  Proof of Lemma 1

PROOF. We need to show that **(1)** $d(u, v) = d(v, u)$, **(2)** $d(u, v) \geq 0$, **(3)** $d(u, v) = 0$ iff $u = v$ and **(4)** $d(u, v) \leq d(u, w) + d(w, v)$. We omit the proofs of first three properties since they are trivial. We prove triangular inequality through contradiction. Let us assume

$$d(u, v) > d(u, w) + d(w, v) \tag{9}$$
$$\text{or, } sp(u, v) + sp(v, u) > sp(u, w) + sp(w, u) + sp(w, v) + sp(v, w)$$

From the definition of shortest paths,

$$sp(u, v) \leq sp(u, w) + sp(w, v) \tag{10}$$
$$\text{and, } sp(v, u) \leq sp(v, w) + sp(w, u)$$

Consequently, Eq. 9 is a contradiction. $\qquad\square$

The exact algorithm to construct an $O(\log^2 |\mathcal{V}|)$-dimensional Lipschitz embedding of $O(\log |\mathcal{V}|)$ distortion is described in [13].

## D  NEUROMLR Training Algorithm

---

**Algorithm 3:** Training Algorithm

---

**Input** : $\mathcal{G} = (\mathcal{V}, \mathcal{E}, \delta, \tau_t), \mathcal{D}$, hyper-parameters : $L$ (GCN layer), $k$ (traffic representation dimensionality), hidden dimension $d_f$ in GCN, $L^M$ (number of layers in MLP)

**Output :** Parameter set $\Theta$

1  $\forall u \in \mathcal{V}, \ \boldsymbol{h_u^0} \leftarrow Lipschitz(u)$
2  $V \leftarrow$ top-$k$ eigenvectors on raw traffic vectors
3  **while** *Training is true* **do**
4   **for** $l = [1, \cdots, L]$ **do**
5    **for** $\forall u \in \mathcal{V}$ **do**
6     $\boldsymbol{h_u^l} \leftarrow \sigma \left( \boldsymbol{W_l} \sum_{v \in N(u) \cup u} \frac{\boldsymbol{h_v^{l-1}}}{\sqrt{(|N(v)|+1)(|N(u)|+1)}} \right)$
7   $Loss(\Theta) \leftarrow 0$
8   $\mathcal{B} \leftarrow minibatch \ from \ \mathcal{D}$
9   **for** $\forall \mathcal{R} \in \mathcal{B}$ **do**
10   $d \leftarrow$ last node in $\mathcal{R}$
11   $t \leftarrow$ start time of$\mathcal{R}$
12   **for** $i \in [1, \cdots, |\mathcal{R}|]$ **do**
13    $curr \leftarrow \mathcal{R}.v_i$
14    $true_{NextNode} \leftarrow \mathcal{R}.v_{i+1}$
15    $\boldsymbol{z_{curr}} \leftarrow \boldsymbol{h_{curr}^L}$
16    $\boldsymbol{z_d} \leftarrow \boldsymbol{h_d^L}$
17    $\boldsymbol{z_t} \leftarrow \boldsymbol{r_t} \boldsymbol{V}$
18    **for** $\forall v \in N(curr)$ **do**
19     $\boldsymbol{z_v} \leftarrow \boldsymbol{h_v^L}$
20     $z \leftarrow \text{Concatenate}(\boldsymbol{z_{curr}}, \ \boldsymbol{z_v}, \ \boldsymbol{z_d}, \ \boldsymbol{z_t})$
21     $\boldsymbol{g_0} \leftarrow z$
22     **for** $l = [1, \cdots, L^M]$ **do**
23      $\boldsymbol{g_l} \leftarrow ReLU\left(\boldsymbol{W_l^M} \boldsymbol{g_{l-1}} + \boldsymbol{b_l^M}\right)$
24     $f(v, curr, d, t) \leftarrow \boldsymbol{g_{L^M}}$
25    $Q((curr, \ true_{NextNode})|curr, d, t; \Theta) \leftarrow \frac{exp(f(true_{NextNode}, \ curr, \ d, \ t))}{\sum_{v' \in N(curr)} exp(f(v', curr, \ d, \ t))}$
26    $Loss(\Theta) \leftarrow Loss(\Theta) + log \ Q((curr, \ true_{NextNode})|curr, d, t; \Theta)$
27  $Loss_{\mathcal{B}}(\Theta) \leftarrow \frac{Loss(\Theta)}{|\mathcal{B}|}$
28  $\Theta \leftarrow Adam(\Theta, \nabla_\Theta Loss_{\mathcal{B}}(\Theta))$

---

# E Computational Complexity analysis of NEUROMLR

We extract the node embeddings $z_{curr}$, $z_d$, and $z_v$ for each neighbor $v \in N(curr)$. The node embeddings are pre-computed and stored. The traffic representation at any time $t$ is independent of the query and is thus maintained separately. Consequently, all these embeddings can be fetched in $O(g)$ time, where $g$ is the average degree in the road network. Computing the transition probability now requires a forward pass through the MLP module for each neighbor of $curr$.

If $d_f$ and $k$ are the dimensions of the node and traffic representations respectively, then the input to the MLP module has size in $O(d_f + k)$. Let the hidden dimension be $d_h$. In the MLP, $L_M$ linear transformations are applied, with intermediate point-wise non linearities; $L_M$ is the number of layers in the MLP. The first and last linear transformation take $O((d_f + k) \times d_h)$ and $O(d_h)$ time respectively. Each intermediate linear transformation consumes $O((d_h)^2)$ time. Assuming $d_h$ and $k$ to be in $O(d_f)$, the total complexity of the MLP module is $O\left(L_M \times d_f^2\right)$.

The total time complexity of predicting $Pr(e = (curr, v) \mid curr, d, t)$ is therefore $O(g \times L_M \times d_f^2)$.

For the pseudocode, refer to Alg 2. Recall that our model estimates $Pr(e|curr, d, t)$ as $Q(e|curr, d, t; \Theta)$.

# F System Configuration Details

All experiments are performed on a machine running Intel Xeon E5-2698v4 processor with $40$ cores, having $1$ Nvidia Tesla V100-DGXS GPU card with 32GB GPU memory, and 256 GB RAM with Ubuntu 16.04.

# G Parameters

The default parameter values for NEUROMLR are summarized in Table 4.

| Parameter | Value |
|---|---|
| $\Delta$ in Alg. 2 | 2 |
| Number of eigenvectors for traffic representation $k$ | 10 |
| Hidden dimension $d_h$ | 128 |
| Number of convolutional layers in GCN | 2 |
| Number of MLP layers | 3 |
| Learning Rate | $10^{-3}$ |

Table 4: Default parameter values in NEUROMLR.

# H Frequency Distribution of Edges

The frequency distribution of edges is shown in Figs. 6a-6c.

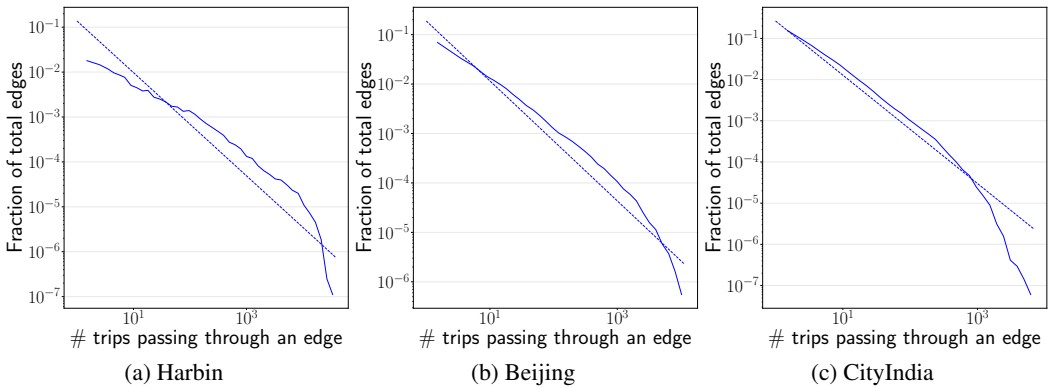

Figure 6: (a-c) Frequency distribution of edge popularity

# I Impact of Trip Length

In Fig. 7, we study the impact of trip length on performance of various algorithms on the Harbin and CityIndia datasets.

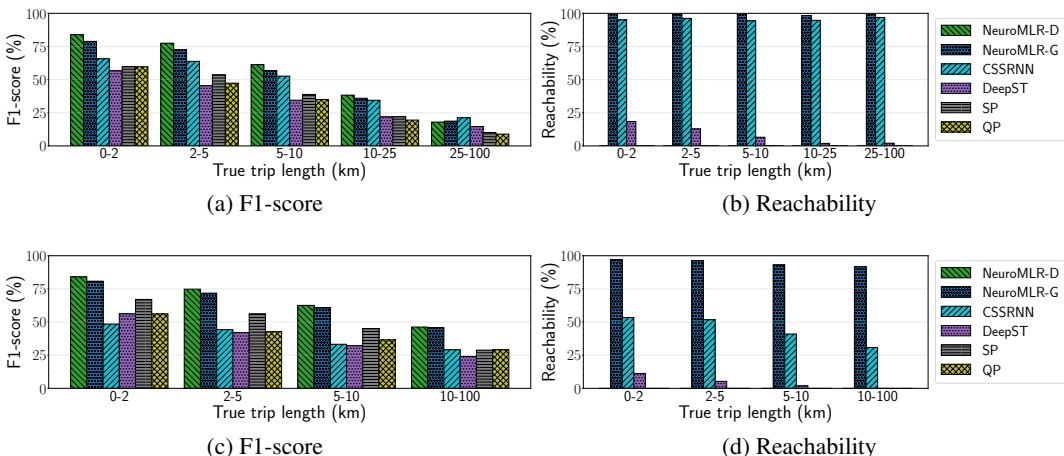

Figure 7: Impact of trip length on accuracy and reachability. **(a,b)** Harbin dataset **(c,d)** CityIndia dataset. The bins for CityIndia were altered since food deliveries operate only for a certain distance radius around each restaurant, causing a low percentage of long trips.

# J Ablation study

**Impact of GCN and Lipschitz Embeddings:** In Figures 8a-8d, we quantify this impact using two more datasets.

**Impact of traffic:** We study the impact of removing the traffic component on NEUROMLR-Greedy's performance. As visible in Table. 5, there is minimal change in performance. Similar results were observed for all datasets. Although this result appears surprising, similar trends have been reported in the literature [10]. This indicates that although traffic may alter speed, they do not significantly alter the transition distributions.

| | Precision (%) | Recall (%) | Reachability (%) | Reachability Distance (km) |
|---|---|---|---|---|
| **Traffic** | 75.6 | 74.5 | 99.1 | 0.013 |
| **W/O Traffic** | 75.4 | 74.5 | 99.2 | 0.016 |

Table 5: Impact of traffic information on NEUROMLR-Greedy's performance in Beijing dataset.

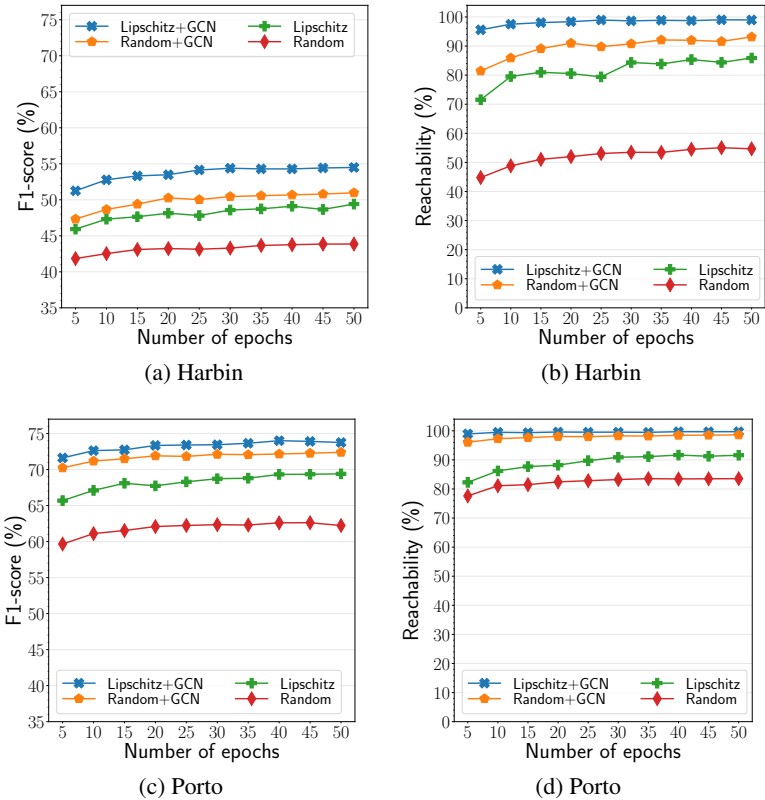

(a) Harbin

(b) Harbin

(c) Porto

(d) Porto

Figure 8: Impact of Lipschitz embeddings and GCN on accuracy and performance of NEUROMLR-Greedy in **(a,b)** Harbin dataset and **(c,d)** Porto dataset.

## K  Impact of Parameters

**Size of hidden dimension:** In Fig. 9a, we measure how the size of the hidden dimension impacts the performance of NEUROMLR-Greedy. Generally, with larger dimensions, we expect the performance to improve as it empowers the model with a larger set of parameters. This trend is visible in Fig. 9a, where the accuracy improves dramatically till 32, and saturates at 128. This result motivates us to set 128 as the default size of hidden dimensions.

**Impact of hidden layers in GCN:** We next examine how the number of convolutional layers in GCN impacts the performance, Fig. 9b presents the results. We observe that the the performance improves till 2 layers, after which it starts to deteriorate. The number of layers determine the extent of information diffusion in the road network. When the number of layers is $L$, the embedding of a node is impacted by those that are up to $L$-hops away. While sharing of information is needed to generalize to unseen nodes, the information may become noisy when it is received from nodes that are very far away. Consequently, we see the pattern visible in Fig. 9b.

**Impact of hidden layers in MLP:** In general, a greater number of hidden layers in MLP is expected to improve performance due to an increase in the capacity of the model. This trend is visible in Fig. 9c.

## L  Experiments on remaining datasets

Table 6 presents the performance of the various algorithms across the two smaller datasets, Chengdu (**CHG**) and Porto (**PT**). Both versions of NEUROMLR continue to outperform all baseline algorithms in both prediction accuracy and reachability. The Porto dataset does not contain time information. Hence, all experiments on Porto ignore the traffic component.

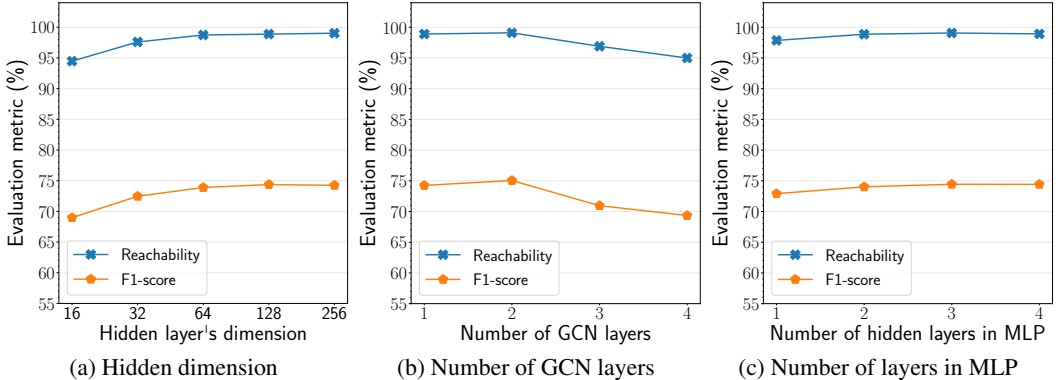

| (a) Hidden dimension | (b) Number of GCN layers | (c) Number of layers in MLP |

Figure 9: Impact of **(a)** hidden dimension size, **(b)** number of GCN layers and **(c)** number of MLP hidden layers on the performance of NEUROMLR-Greedy. All plots in this figure use the Beijing dataset.

| Algorithm | Precision (%) | | Recall (%) | | Reachability (%) | | Reachability Distance(km) | |
|---|---|---|---|---|---|---|---|---|
| | CHG | PT | CHG | PT | CHG | PT | CHG | PT |
| **NEUROMLR-D** | **86.7** | **79.2** | **84.2** | **70.9** | — | — | — | — |
| **NEUROMLR-G** | 86.1 | 77.3 | 83.8 | 70.7 | **99.9** | **99.6** | **0.0002** | **0.001** |
| **CSSRNN** | 84.5 | 70.6 | 83.2 | 71.5 | 99.4 | 98.1 | 0.009 | 0.046 |
| **DEEPST** | 52.9 | 55.3 | 41.6 | 30.3 | 13.8 | 7.0 | 1.023 | 1.211 |
| **SP** | 60.3 | 58.2 | 55.1 | 47.7 | — | — | — | — |
| **QP** | 57.7 | — | 54.7 | — | — | — | — | — |

Table 6: Comparison of NEUROMLR against the benchmarked algorithms on the four different metrics of average precision, average recall, percentage of trips that reached destination, and average distance from the true destination. The best performance for each dataset is highlighted in bold. QP is undefined in Porto since this dataset does not contain timestamps.

## M   Performance of DEEPST

**Deviation from results in DEEPST [10]**

We observe the following differences. **(1)** The performance of CSSRNN is significantly better than reported in [10]. **(2)** The performance of DEEPST is weaker in our experiments than reported in [10], **(3)** CSSRNN performs better than DEEPST in our evaluation.

While it is hard to pin-point the exact reasons, one possible reason could be that the map-matching algorithms used are different. We note that although the raw datasets of Chengdu and Harbin are the same, the mean number of road segments in our map-matched datasets is close to two times more than in DEEPST (14 vs. 23 in Chengdu and 24 vs. 57 in Harbin).

**Poor reachability of DEEPST**

In order to effectively share the statistical strength across trips, DEEPST adopts a clustering-inspired approach where where all nodes are partitioned into $K$ disjoint groups; $K$ being a hyperparameter. Now, consider queries $q_1 : \langle s, d_1, t \rangle$ and $q_2 : \langle s, d_2, t \rangle$. Although $q_1$ and $q_2$ are different due to two different destinations, they may map to the same input parameters for the neural architecture if $d_1$ and $d_2$ share the same group. Consequently, although DEEPST guides the recommended route in the correct direction, it rarely reaches the destination.

## N   Attention module

The importance of each of the input factors - current node $curr$, destination $d$, traffic condition at time $t$, and the transition node $v$ can be learnt through an attention module. To highlight the importance of

attention, consider a node where the transition probabilities are invariant to traffic conditions. Here, the attention of $z_t$ should be low. Similarly, if there is only one outgoing edge from a node, the importance $z_d$ is low. We mathematically capture these hyper-local intricacies as follows.

We denote $e_{xy}$ as the importance of input $y$ for input $x$.

$$e_{xy} = a(\boldsymbol{z}_x, \boldsymbol{z}_y) \tag{11}$$

The output representation $\widetilde{z}_i \in \mathbb{S} = \{\boldsymbol{z_{curr}}, \ \boldsymbol{z_v}, \ \boldsymbol{z_d}, \ \boldsymbol{z_t}\}$ for each input is a linear combination of all inputs where the weights correspond to these importance scores. The raw importances $e_{xy}$ are normalised using the *softmax* function to obtain final weights $\alpha_{x,y}$.

$$\alpha_{xy} = \frac{\exp(e_{xy})}{\sum_{i \in \mathbb{S}} \exp(e_{xi})} \tag{12}$$

The function $a : \mathbb{R}^{d_f} \times \mathbb{R}^{d_f} \to \mathbb{R}$ is a single-layer feed-forward neural network, which maps the input representations to their relative importance score. Mathematically,

$$e_{xy} = a(\boldsymbol{z}_x, \boldsymbol{z}_y) = ReLU(\boldsymbol{a} \cdot [\boldsymbol{z_x} || \boldsymbol{z_y}])) \tag{13}$$

$a(\boldsymbol{z}_x, \boldsymbol{z}_y)$ is parameterised by a row matrix $\boldsymbol{a} \in \mathbb{R}^{1 \times 2d_f}$. $||$ denotes the *concatenation* operator. $d_f$ denotes the dimension of each input representation $\boldsymbol{z_i} \in \mathbb{S}$. Finally, the attention-weighted representation of each input representation $\boldsymbol{z_x} \in \mathbb{S}$ is computed as:

$$\widetilde{\boldsymbol{z}}_x = \sigma \left( \sum_{\boldsymbol{z_i} \in \mathbb{S}} \alpha_{xi} \cdot \boldsymbol{z}_i \right) \tag{14}$$

where $\sigma$ is the ReLU activation function. The final output from the attention module is the concatenated attention-weighted input vectors.

$$\widetilde{\boldsymbol{z}} = [\widetilde{\boldsymbol{z}}_{curr} || \widetilde{\boldsymbol{z}}_n || \widetilde{\boldsymbol{z}}_d || \widetilde{\boldsymbol{z}}_t] \tag{15}$$

For employing this attention module, $z$ was replaced with $\widetilde{z}$ in line 21 of Alg. 3. Since we did not observe any significant improvement with this change, we performed all our experiments with the concatenation approach (line 20 of Alg. 3).

## O  Inductive Learning Experiments on other Datasets

In Figures 10 and 11, we demonstrate the benefits of inductive learning on two more datasets, CityIndia and Harbin respectively. We investigate how the performance of NEUROMLR and CSSRNN varies with the popularity of source and destination nodes of test trips. Our key observations are highlighted below.

1. NEUROMLR continues to adapt more gracefully in all cases.

2. Similar to the trends observed with the Beijing dataset, the performance of CSSRNN on the **P-U** category trips undergoes a substantial decline.

3. For a food delivery dataset with source nodes (restaurants) being much more frequent than destination nodes (delivery locations), the percentage of trips in **P-U** category is expected to be higher than **U-P** and **U-U** categories. This trend is visible in Fig. 10a. Consequently, the aggregate performance metrics of CSSRNN on CityIndia are poor (Table 2).

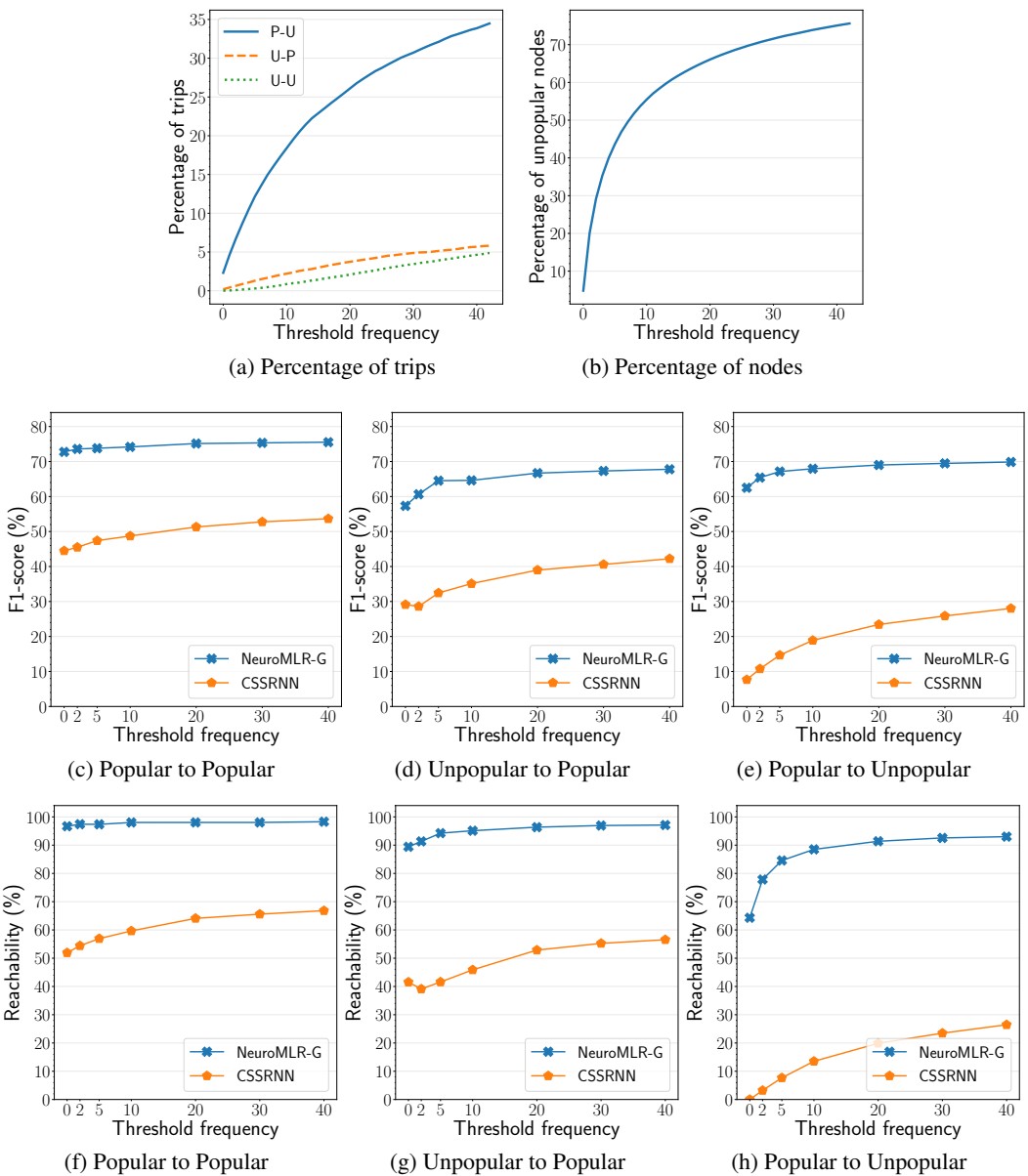

Figure 10: All these experiments were performed on the **CityIndia** dataset. **(a)** Percentage of test data trips in **P-U**, **U-P** and **U-U** categories. **(b)** Percentage of nodes in the road network marked as unpopular as a function of threshold frequency. **(c-h)** Variation of Reachability & F1-score with the threshold frequency for different popularity categories.

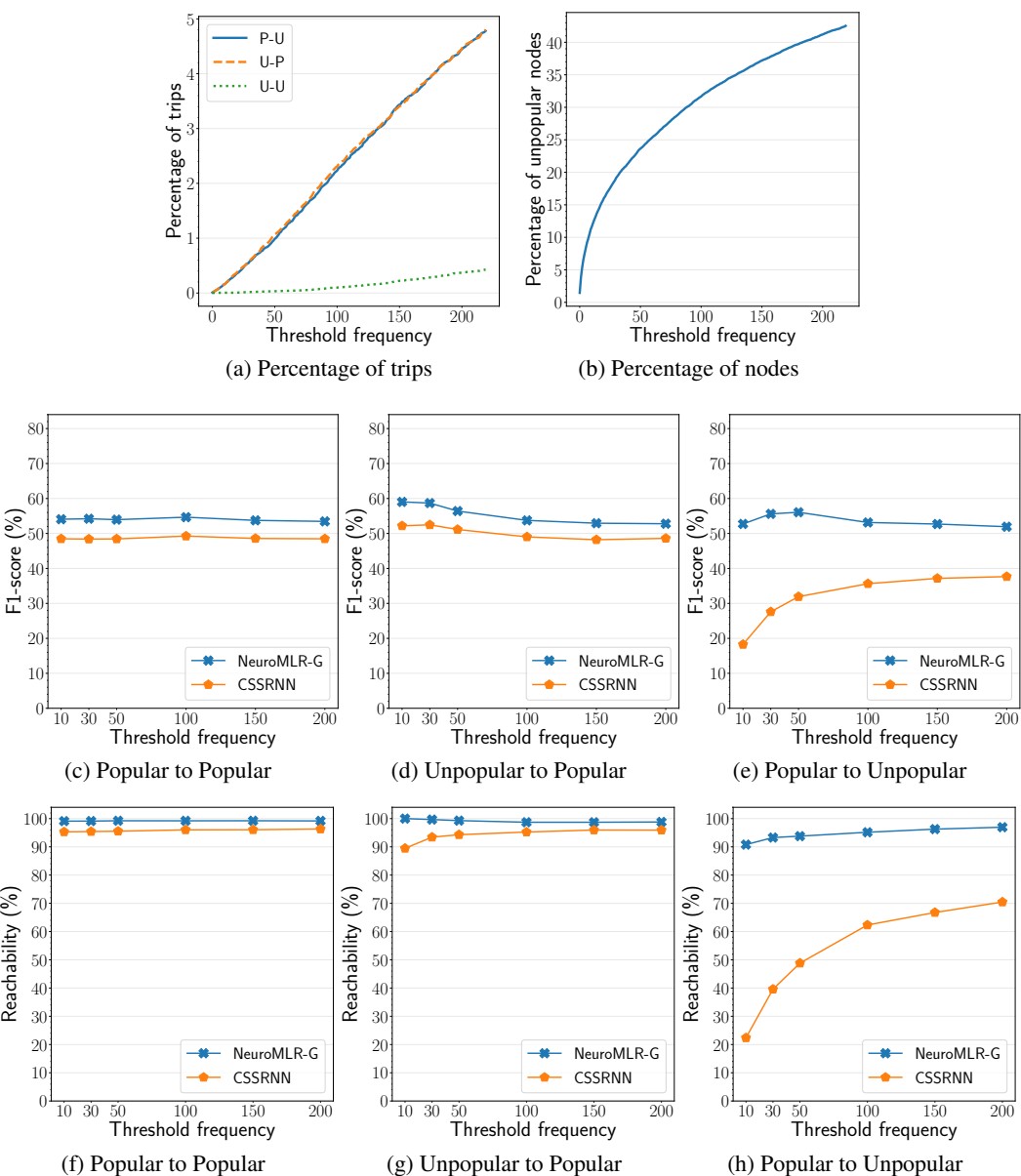

(a) Percentage of trips

(b) Percentage of nodes

(c) Popular to Popular

(d) Unpopular to Popular

(e) Popular to Unpopular

(f) Popular to Popular

(g) Unpopular to Popular

(h) Popular to Unpopular

Figure 11: All these experiments were performed on the **Harbin** dataset. **(a)** Percentage of test data trips in **P-U**, **U-P** and **U-U** categories. Note that the percentages for **P-U** and **U-P** are very similar for all the thresholds and hence the overlapping plots. **(b)** Percentage of nodes marked as unpopular as a function of threshold frequency. **(c-h)** Variation of Reachability & F1-score with the threshold frequency for different popularity categories.

# P    Justification of Low Absolute Performance

In a traditional prediction setup (such as classification or regression), for a specific input, there is only one correct output. In route recommendations that is not the case; for the same input (source-destination pair), we may have different outputs (routes). If the likelihood distribution of the output space for a given input has high entropy, then the problem gets more difficult.

To substantiate this claim with real data, we formulate the idea of a *majority-route*. Given a pair of source and destination nodes $(s, t)$, the majority-route selects the route that is taken most frequently to go from source $s$ to destination $t$ in the trajectory database. The probability of the majority-route is simply

$$\frac{\text{frequency of majority-route for } (s, t)}{\text{number of routes from } s \text{ to } t \text{ in trajectory database}}$$

In Table 7, we plot the *cumulative* distribution of the percentage of source-destination pairs against the probability of the majority-route. For example, in the Harbin row, the entry for column 0.6 indicates that the majority-route of 75.3%. source-destination pairs have a probability of 0.6 or below.

Overall, it is amply clear from the data, that route recommendation is indeed a difficult problem since even the most likely route for a given input often occurs with a low probability.

| City | 0.2 | 0.4 | 0.6 | 0.8 | 1.0 |
|---|---|---|---|---|---|
| **Beijing** | 0.1 | 3.2 | 35.7 | 41.8 | 100 |
| **Chengdu** | 2.2 | 17.9 | 55.5 | 70.4 | 100 |
| **CityIndia** | 0.8 | 9.1 | 60.4 | 67.6 | 100 |
| **Harbin** | 2.8 | 19.0 | 75.3 | 82.7 | 100 |
| **Porto** | 4.1 | 20.7 | 61.3 | 72.8 | 100 |

Table 7: Cumulative distribution of the percentage of source-destination pairs against the probability of the majority-route.