# OpenReview forum: " NeuroMLR: Robust & Reliable Route Recommendation on Road Networks "
_NeurIPS.cc/2021/Conference — NeurIPS 2021 Poster_

### Official Review · Reviewer_F1v9 · 2021-07-12

**Rating:** 6
**Confidence:** 4

**Summary:**

The paper introduces a route recommendation model based on graph neural networks with Lipschitz embedding initialization. The proposed model is compared with several baseline models and the evaluation shows that it outperforms baselines in terms of precision, recall, and reachability. Ablation study is done to demonstrate the importance of both Lipschitz embedding and GCN components. The model which uses inductive learning shows that it significantly outperforms baselines when the source/destination node is unpopular.

**Limitations And Societal Impact:**

I don't see any potential negative societal impact of their work.

**Main Review:**

The model is easy to understand and explained clearly in the paper. There are ablation studies to help understand the impact of each component in the model. The results look promising compared with the baseline in the paper.

However, I feel like there should be more exploration on the model component as GCN is only one possible way of implementing graph neural networks. What if different GNN architectures are used? Have you tried them? Are there any GNN based model for route recommendations? Maybe it's more fair to compare with them? It seems to me that GNN is only used to learn node embeddings and then an MLP is applied to generate the transition probability. Have you tried GNN models that can learn edge embeddings as well and include edge embeddings in the MLP input? Have you tried different ways to learn these embeddings and compare them? It seems that the proposed model mostly stack existing models together so the real novelty is not very significant. It would be better if the paper could explain or explore more on the alternatives, for example, maybe using GNN to also execute graph algorithms (https://openreview.net/forum?id=SkgKO0EtvS) instead of using MLP to generate transition probabilities.

There should be more explanation on why shortest path or quickest path does not perform well. I would assume in the dataset, people use existing apps to navigate which would potentially give a route based on a combination of route distance and time (which may consider live traffic). So probably crudely using shortest/quickest path without the live traffic is why is not performing well.

For Reachability Distance in the evaluation metrics, it seems to me network distance is better than Haversine distance? If two locations are not physically reachable in the road network, it really doesn't matter how close they are in terms of Haversine distance.

In terms of inductively learning, I think a better way to advance it is to handle dynamic network, such as road closures or new road being introduces. This would be more appealing in a practical sense.

**Time Spent Reviewing:**

2

---

> ### Author Response · Authors · 2021-08-10
> **Response to Reviewer 3 (F1v9)**
>
> **Comment 1: I feel like there should be more exploration on the model component as GCN is only one possible way of implementing graph neural networks. What if different GNN architectures are used? Have you tried them? Are there any GNN based model for route recommendations? Maybe it's more fair to compare with them? It seems to me that GNN is only used to learn node embeddings and then an MLP is applied to generate the transition probability. Have you tried GNN models that can learn edge embeddings as well and include edge embeddings in the MLP input? Have you tried different ways to learn these embeddings and compare them? It seems that the proposed model mostly stack existing models together so the real novelty is not very significant. It would be better if the paper could explain or explore more on the alternatives, for example, maybe using GNN to also execute graph algorithms (https://openreview.net/forum?id=SkgKO0EtvS) instead of using MLP to generate transition probabilities.**
>
> *Answer:* We break up our response into the following categories:
>
> 1. Justification for GCN
> 2. Can https://openreview.net/forum?id=SkgKO0EtvS be used for our problem?
> 3. Route recommendation algorithms using GNN
> 4. Use of edge-embeddings for prediction
> 5. Novelty
>
> ### 1. Justification for GCN
>
> **Why GCN?** The key task of GCN is to enable model inductivity. Consider a node that is never traversed by any of the training trajectories. The initial embedding of such a node would never be updated during training. Furthermore, even if a node is seen, but infrequently, its embedding will be noisy due to a lack of training data. GCN circumvents this problem by sharing information across neighbourhood nodes. We use a mean-pool in our convolution operation, which is like applying a smoothing filter. Thus, even if a node is not seen much, but its neighbour is seen frequently, GCN will ensure that the two nodes have similar embeddings.
>
> **Alternatives:** Several other GNNs also utilize mean-pool convolution filters (Ex. GraphSage and Graph Attention Network (GAT)). While the difference between GCN and GraphSage are minor, GAT utilizes an attention-weighted mean pool. Below, we present an ablation study, where we replace GCN with GraphSage and GAT. The results are similar since they impart a similar inductive bias on node embeddings.
>
> #### Beijing
>
> |Metric|GCN|GAT|GraphSAGE|
> |-|-|-|-|
> Precision|**75.6**|74.9|75.4|
> Recall|**74.5**|73.9|74.2|
> Reachability|**99.1**|**99.1**|99.0|
>
> #### Chengdu
>
> |Metric|GCN|GAT|GraphSAGE|
> |-|-|-|-|
> Precision|**86.1**|85.7|85.8|
> Recall|**83.8**|83.4|83.5|
> Reachability |**99.9**|**99.9**|**99.9**|
>
> **GCN Vs. MLP:**
> Generally, the final layer of most GNNs is an MLP including the paper referred to in the comment (https://openreview.net/forum?id=SkgKO0EtvS). Thus, the GCN and the MLP jointly compute the transition probabilities. The importance of GCN towards accurate prediction is empirically established in Fig. 3.
>
> ### 2. Can https://openreview.net/forum?id=SkgKO0EtvS be used for our problem?
>
> *Answer:* The referred paper utilizes a message-passing GNN to learn imitation of multiple algorithms jointly. This makes it unsuitable for our problem due to the following reasons:
>
> 1. Our goal is to learn a *generative model* for trajectories. In contrast, the referred paper learns to imitate a search algorithm (such as BFS, Bellman-Ford, or Prim's). The transition steps of these algorithms are deterministic. Thus, the primary task is to learn the rules that govern these transitions. In our case, the transitions are governed by an underlying hidden distribution and our goal is to learn this distribution.
>
> 2. The referred paper focuses on *multi-task learning*. Specifically, given multiple algorithms that share some structure in their exploration strategies, the authors show that learning their imitations jointly yields better performance than learning them individually. In our problem, we do not have the notion of multi-task learning.
>
> ### 3. Route recommendation algorithms using GNN
> **Route Recommendations:** CSSRNN and DeepST are the state of the art algorithms for route-recommendation. However, neither of them use GNNs. Citation 20 in our paper used Graph Attention Networks for personalized route-recommendations. As mentioned in Section 5.1, this algorithm is not compatible with our problem due to the personalization aspect. We did not use GAT since, as shown above, it did not improve accuracy.
>
> **Road Network Analysis:** In the broader space of road network analysis, [#1] learns embeddings of nodes for speed estimation of vehicles. [#2] learns general-purpose node embeddings on road networks for tasks such as node classification and destination prediction, etc.
>
> [#1] Tobias et al. Graph Convolutional Networks for Road Networks. In SIGSPATIAL 2019.
>
> [#2] Ning et al. Learning Effective Road Network Representation with Hierarchical Graph Neural Networks. KDD 2020.
>
> ### 4. Use of edge-embeddings for prediction
> We did try edge-embeddings. Specifically, we constructed a dual graph $G_d$ from the original road network graph $G$, where each edge in $G$ corresponds to a node in $G_d$, and two nodes in $G_d$ are connected if their corresponding edges in $G$ share an endpoint. On this dual graph, we used NeuroMLR. This version is also part of our released codebase. Compared to running on $G$, $G_d$ obtained minor improvements. However, it also increased training time ($G_d$ is a significantly larger graph than $G$), inference time as well as memory consumption. We will mention this information in our experiments section.
>
> ### 5. Novelty
>
> *Answer:* Please refer to Reviewer 2(ABKf) Comment 1 for the discussion on novelty.
>
> **Comment 2: There should be more explanation on why the shortest path or quickest path does not perform well.**
>
> *Answer:* We had briefly touched upon this question in the first para of Section 1.1. We present a more detailed discussion here.
>
> Several studies have monitored automobile movement patterns and have shown that people rarely travel in the shortest or quickest paths [\#1, \#2]. In [\#1], trajectories of 143 people were tracked using GPS over a three-week period. In a total of 25,157 trips collected in this study, two-thirds of the trips did not travel in the quickest paths. Furthermore, shortest paths were never travelled in unless the shortest path coincided with the quickest path. In [\#2], a similar study from Nagoya, Japan draws the same conclusion.
>
> It has been shown that automobile travellers select routes based on several latent factors such as the presence of tolls [\#2,\#4], travel time reliability [\#3], traffic lights [\#5], and road aesthetics [\#6,\#7]. Furthermore, it has also been shown that human beings systematically misperceive travel time [\#8]
>
> Owing to these numerous factors, the accuracies of the shortest and quickest paths are low. The weak performance of shortest/quickest paths is consistent with the results reported in CSSRNN [22] and DeepST [9]. We will now add a more detailed discussion.
>
> [\#1] Zhu et al. (2015) Do People Use the Shortest Path? An Empirical Test of Wardrop’s First Principle. PLoS ONE 10(8)
>
> [\#2] Morikawa et al. Simulation Approaches in Transportation Analysis Springer Science+ Business Media, Inc, New York. 2005;135:136.
>
> [\#3] Carrion et al. Value of travel time reliability: A review of current evidence. Transportation research part A: policy and practice. 2012;46(4):720–741.
>
> [\#4] Papinski et al. Exploring the route choice decision-making process: A comparison of planned and observed routes obtained using person-based GPS. Transportation research part F: traffic psychology and behaviour. 2009;12(4):347–358.
>
> [\#5] Palat et al. Numerosity heuristic in route choice based on the presence of traffic lights. Transportation Research Part F: Traffic Psychology and Behaviour. 2014;22:104–112.
>
> [\#6] Daniele Quercia et al. The shortest path to happiness: Recommending beautiful, quiet, and happy routes in the city. In Proceedings of the 25th ACM Conference on Hypertext and Social Media, page 116–125, 2014.
>
> [\#7] Zhang L et al. Determinants of route choice and value of traveler information: a field experiment. Transportation Research Record: Journal of the Transportation Research Board. 2008;2086(1):81–92.
>
> [\#8] Parthasarathi P et al. Network structure and travel time perception. PloS one. 2013;8(10):e77718. pmid:24204932
>
> **Comment 3: For Reachability Distance in the evaluation metrics, it seems to me network distance is better than Haversine distance?**
>
> *Answer:* We agree that road network distance is a better measure. We used Haversine since (1) Typically, Haversine and road network distances are correlated. (2) Haversine is a lower-bound on the road network distance. (3) Computing Haversine is significantly faster than computing shortest paths.
>
> Nonetheless, we have re-computed the reachability distance of NeuroMLR-Greedy based on road network distance. The table below presents the results.
>
> |Distance (in Km)|Beijing | Chengdu|CityIndia|Harbin|Porto|
> |-|-|-|--|-|-|
> Haversine|0.01|0.0002|0.03|0.02|0.001
> Road Network|0.02|0.0002|0.04|0.02|0.005
>
> As visible, the distances are highly similar. Furthermore, the road network reachability distances of NeuroMLR are significantly smaller than the Haversine reachability distances of CSSRNN and DeepST (See Table 2 in the main paper). Since Haversine is a lower bound, we can guarantee that the road network reachability distances of NeuroMLR will be significantly smaller than the road network reachability distances of CSSRNN and DeepST.
>
>
>
> **Comment 4: In terms of inductively learning, I think a better way to advance it is to handle dynamic networks, such as road closures or new roads being introduced. This would be more appealing in a practical sense.**
>
> *Answer:* We have conducted this experiment in our response to Reviewer 1(egXq) Comment 3. Please refer to this response for the related discussion.

---

### Official Review · Reviewer_ABKf · 2021-07-13

**Rating:** 6
**Confidence:** 2

**Summary:**

This paper proposes a method for recommending travel routes in a city. As shown in Figure 2, in order to consider the meaningful representation of road network, this paper uses a combination of Graph Convolutional Network with Lipschitz embeddings. Multi-layered Perceptron (MLP) is then applied to predict the next visit.

**Limitations And Societal Impact:**

Limitation of the proposed method is mentioned in the conclusion. The negative impacts of this study were not mentioned in the paper, nor were there any points that bothered me when I read the paper.

**Main Review:**

- Strengths
  - This paper is well-structured and easy to follow.
  - The proposed architecture seems reasonable for this task. Through the ablation study (Sec. 5.4), it has also been shown that each component of the proposed method (Lipschitz embeddings and GCN) is working effectively.
  - The minimum necessary comparisons with two conventional methods have been made using multiple data sets.
- Weaknesses
  - The novelty of the proposed method should be described in more detail. Is the key point of this work is to devise a pre-processing (PCA and Lipschitz embedding)? Does the majority of the proposed architecture relies on existing methods?
  - I am concerning about whether the comparison experiment was done under fair conditions. (1) The authors set a restriction of 36 hours for the parameter training time, but has this restriction affected the experimental results? (e.g., would the same conclusion be reached if the time were 72 hours?). (2) For both DEEPST and CSSRNN, the authors use the default parameters prescribed by these papers. However, unless the parameters of the existing methods are learned in a way that is tailored to the current data, it may not be a fair comparison between the proposed method and the conventional method. (3) Why did the authors use only one of three datasets for the experiments on "Accuracy" and "Reachability"? I am also curious about the large difference in performance between each dataset (each town).

**Time Spent Reviewing:**

3 hours

---

> ### Author Response · Authors · 2021-08-10
> **Response to Reviewer 2 (ABKf)**
>
> **Comment 1: The novelty of the proposed method should be described in more detail. Is the key point of this work is to devise a pre-processing (PCA and Lipschitz embedding)? Does the majority of the proposed architecture rely on existing methods?**
>
>
> *Answer:*  The novel components of our work are as follows:
>
> * Our contribution lies not in the design of the individual components (Ex. Lipschitz embeddings, GCN, PCA, etc.), but the unique way in which they are combined to design an effective and efficient route recommendation algorithm. On the whole, this enables *inductive* modelling of trajectory data, i.e., the ability to predict on unseen source-destination pairs. This results in significantly more accurate performance on unseen or lesser-seen nodes (See Figs. 4c-4i) when compared to the baselines. Furthermore, NeuroMLR learns more effectively from smaller training data (Fig. 4a-4b).
>
> * None of the existing learning-based algorithms guarantee reachability. Reachability is an extremely important metric since if the recommended route does not reach the intended destination, it severely compromises the end-user experience. NeuroMLR carefully decouples the two problems of modelling transition probabilities and route search. This allows us to employ Dijkstra's algorithm in the route search component, which *guarantees* reachability.
>
>
> **Comment 2: I am concerning about whether the comparison experiment was done under fair conditions. (1) The authors set a restriction of 36 hours for the parameter training time, but has this restriction affected the experimental results? (e.g., would the same conclusion be reached if the time were 72 hours?). (2) For both DEEPST and CSSRNN, the authors use the default parameters prescribed by these papers. However, unless the parameters of the existing methods are learned in a way that is tailored to the current data, it may not be a fair comparison between the proposed method and the conventional method. (3) Why did the authors use only one of three datasets for the experiments on "Accuracy" and "Reachability"? I am also curious about the large difference in performance between each dataset (each town).**
>
> *Answer:* We break up our answer into (1) Model convergence, (2) Parameters, and (3) Dataset selection, and (4) Difference in performance across cities.
>
> ### 1. Model Convergence
>
> We trained all models till convergence of validation loss. Since all models converged within 36 hours, we wrote this statement. We agree our statement should have been more precise. We will change this to explicitly mention that all models are trained till convergence of validation loss. We will also add the convergence data of CSSRNN and DeepST in the supplementary.
>
> To support our statement with empirical data, we show the validation loss of CSSRNN and DeepST against the number of epochs. "-" indicates that execution was terminated due to convergence. As visible, all models converged in terms of validation loss.
>
> ### CSSRNN
>
> |Epochs | 20 | 40 | 60 | 80 | 100 | 120 | 140 | 160 | 180 | 200 | 220 | 240 | 260 | 280 |
> |----------|-----|------|----|------|-------|------|-------|-------|------|------|--------|-------|-------|------|
> Beijing|10.18  | 6.20 | 7.20| 8.38| 9.61|-|-|-|-|-|-|-|-|-|
> Chengdu | 6.44  |  2.56 |  2.51  |  2.49  | 2.48  | 2.49  | 2.50|-|-|-|-|-|-|-|
> CityIndia| 16.32 | 14.66 | 14.04 | 13.65 | 13.41 | 13.22 | 13.12 | 13.05 | 13.04| 13.07 | 13.16| 13.23 | 13.39 | 13.54 |
> Porto | 8.65| 5.77| 5.46| 5.39 |5.43|5.50|5.57|5.70|5.81|5.91|-|-|-|-|
>
>
> ### DeepST
>
> |Epochs | 20 | 40 | 60 | 80 | 100 | 120 | 140 | 160 | 180 | 200 | 220 | 240 | 260 | 280 |
> |----------|-----|------|----|------|-------|------|-------|-------|------|------|--------|-------|-------|------|
> Beijing| 61.75 |43.18 |35.41 | 32.00|31.11|30.32| 29.93| 30.12|30.63|31.78|-|-|-|-|
> Chengdu| 23.41 |21.25 |20.39 |20.20|20.13|20.23|20.34|20.59||-|-|-|-|-|
> CityIndia|72.07  | 36.86 | 26.87| 23.79| 21.41 |20.21 |19.57| 19.27| 19.10 |19.11 |19.38|20.06|-|-|-|
> Porto | 19.52 |15.37 |13.51 | 12.82 | 12.56| 12.33| 12.49|12.88|-|-|-|-|-|-|
>
>
> ### 2. Parameters
>
> We apologize again for not being precise enough. We did study the hyperparameters of both CSSRNN and DeepST against the datasets and chose the values where validation loss is minimized. These parameters coincided with the default parameters mentioned in the papers. This result is not surprising since several of the datasets are common to CSSRNN and DeepST (Porto, Chengdu, and Harbin). We will make our parameter selection policy explicitly clear.
>
>
> ### 3. Dataset Selection:
>
> * **Accuracy and Reachability:** We have reported the Accuracy and Reachability on all datasets in our submission. Specifically, Table 2 presents the accuracy and reachability numbers in Beijing, Harbin, and CityIndia. The same metrics for Porto and Chengdu are provided in Table 6 in the Supplementary. The reference to Table 6 is noted as a footnote on Page 7 of the main paper.
>
> * **Ablation Study:** The main paper presents the ablation study only on Beijing due to space limitations. As noted in Sec 5.4, the same experiment on other datasets is provided in Appendix L of Supplementary.
>
> * **Inductive Learning:** In Sec 5.3, we present the inductive learning benefits on the Beijing dataset. The same experiments on Harbin and CityIndia are provided in Figures 9 and 10 in the Supplementary.
>
> ### 4. Difference in performance across cities
>
> *Answer:* There are two factors that play an important role in determining the accuracy: (1) Variability in Output Space, and (2) Volume of training data.
>
> #### **(1) Variability in Output Space**
>
> Reviewer 1(egXq) has kindly provided an accurate explanation on why precision-recall numbers are low in route recommendations in his/her W1 (Comment 0). Specifically, in a traditional prediction setup (such as classification or regression), for a specific input, there is only one correct output. In route recommendations that is not the case; for the same input (source-destination pair), we may have different outputs (routes). If the likelihood distribution of the output space for a given input has high entropy, then the problem gets more difficult. The variation in accuracy numbers across cities can be explained with this likelihood.
>
> To elaborate, first, we formulate the idea of a *majority-route*. Given a pair of source and destination nodes $(s,t)$, the majority-route selects the route that is taken most frequently to go from source $s$ to destination $t$ in the trajectory database. The probability of the majority-route is simply $\frac{\text{frequency of majority-route for } (s,t)}{\text{All seen routes from $s$ to $t$ in trajectory database}}$.
>
> In the table below, we plot the *cumulative* distribution of the percentage of source-destination pairs against the probability of the majority-route. For example, in the Harbin row, the entry for column 0.6 indicates that the majority-route of 75.3% source-destination pairs have a probability of 0.6 or below. Thus, the quicker the cumulative distribution approaches 100%, the more difficult is the dataset.  We clearly see that Harbin has the steepest rise towards 100. This also coincides with the fact that the F-Score in Harbin is the lowest (See the F-score Table below). Porto has the second-most steepest rise, and the second-lowest F-score is also seen in this city.
>
> **Cumulative distribution of the likelihood of the majority-route**
>
> |City| 0.2 | 0.4 | 0.6 | 0.8 | 1.0|
> |----|-----|-----|-----|-----|----|
> Beijing | 0.1 |  3.2 | 35.7  | 41.8 |  100
> Chengdu | 2.2 | 17.9 | 55.5 | 70.4 | 100
> CityIndia|0.8|9.1|60.4|67.6|100
> Harbin | 2.8 | 19.0 | 75.3 | 82.7 | 100
> Porto | 4.1 |20.7 |  61.3 | 72.8| 100
>
>
> **F-score achieved by NeuroMLR in each dataset**
>
> |City| F-score|
> |----|--------|
> Beijing| 77.19
> Chengdu| 85.43
> City India| 75.42
> Harbin| 56.67
> Porto | 74.80
>
>
> #### **(2) The volume of training data:**
>
> Based on the above analysis, we notice one aberration. Although Beijing has a much milder rise towards 100%, its F-score is only marginally better than Porto. Furthermore, Chengdu has the highest F-score although its cumulative distribution of majority-route likelihood is steeper than Beijing. These aberrations can be explained by considering the volume of training data available for each dataset. Specifically, as noted in Table 1, Chengdu has the highest number of training trajectories. Hence, this leads to the best performance. The performance in Beijing is not as strong as in Chengdu since the road network of Beijing is $\approx 8$ times larger, and yet it has a $\approx 7$ times lower number of trajectories.
>
> We will include a summary of the above discussion in the paper along with a pointer to the detailed discussion and statistics in the supplementary.

---

### Official Review · Reviewer_egXq · 2021-07-15

**Rating:** 8
**Confidence:** 3

**Summary:**

This paper presents a new approach for route recommendation. The key idea of the approach is to model the mobility pattern as a Markov process and learn the local transition probability from historical data through a GCN with Lipschitz embedding. The proposed approach overcomes two limitations in the existing works - (1) limited reachability and (2) limited generalizability on unseen/lesser-seen nodes. The paper evaluates their approach on public datasets with a few baselines. The improvement is significant and there are also a good amount of ablation studies and case studies to help the readers better understand the effectiveness of the proposed solution.

**Limitations And Societal Impact:**

No potential negative societal impact.

**Main Review:**

Strengths:

S1 The idea of this paper has its novelty in route recommendation. The way it models the route recommendation problem can inspire future works.

S2 The paper is well written. It is easy to follow and contains sufficient details.

S3 The experiments are sufficient and well-designed.


Weaknesses:

W1 The precision and recall seem low but this may not be an issue of the proposed approach but the metric itself. There might be several reasonable routes going from A to B, and not all the drivers will follow the same route. By this nature, there should be an upper bound of the accuracy. For example, if there are 2 reasonable routes from A to B, and 80% of the drivers take route-1 (most likely route) and 20% of the drivers take route-2 (assume the traffic conditions and all other factors are the same). In this case, you may only be able to achieve 80% precision if the routing queries are all anonymous. Given the existence of this upper bound, the accuracy numbers reported in the paper might be good enough.


Additional Comments:

C1 I wonder if there is a way to estimate the upper bound (see W1) of the accuracy from the data? Showing this estimation can be helpful to understand the nature of the most likely route prediction problem.

C2 At line 254, training all the models for up to 36 hours does not seem to be a fair comparison. It would be better to fix the training epochs rather than training time.

C3 How does the proposed approach handle changes in the road network? For example, there could be temporary road closure caused by construction or crashes.

C4 At line 104, should it be Prob. 1 rather than Prob. 2?





**Time Spent Reviewing:**

5

---

> ### Author Response · Authors · 2021-08-10
> **Response to Reviewer 1 (egXq)**
>
> **Comment 0 (W1): The precision and recall seem low but this may not be an issue of the proposed approach but the metric itself. There might be several reasonable routes going from A to B, and not all the drivers will follow the same route. By this nature, there should be an upper bound of the accuracy. For example, if there are 2 reasonable routes from A to B, and 80% of the drivers take route-1 (most likely route) and 20% of the drivers take route-2 (assume the traffic conditions and all other factors are the same). In this case, you may only be able to achieve 80% precision if the routing queries are all anonymous. Given the existence of this upper bound, the accuracy numbers reported in the paper might be good enough.**
>
> *Answer:* This observation is indeed correct and one of the key reasons why precision-recall numbers in route recommendation algorithms are not very high. This trend is consistent with results reported in CSSRNN [22] and DeepST [9]. At its core, the problem arises from the fact that for the same input (source-destination pair), we may have different outputs (routes). If the likelihood distribution of the output space for a given input has high entropy, then the problem gets more difficult.
>
> To substantiate this claim with real data, we formulate the idea of a *majority-route*. Given a pair of source and destination nodes $(s,t)$, the majority-route selects the route that is taken most frequently to go from source $s$ to destination $t$ in the trajectory database. The probability of the majority-route is simply $\frac{\text{frequency of majority-route for } (s,t)}{\text{All seen routes from $s$ to $t$ in trajectory database}}$.
>
> In the table below, we plot the *cumulative* distribution of the number of source-destination pairs against the probability of the majority-route. For example, in the Harbin row, the entry for column 0.6 indicates that the majority-route of 75.3% of the source-destination pairs have a probability of 0.6 or below.
>
>
> |City| 0.2 | 0.4 | 0.6 | 0.8 | 1.0|
> |----|-----|-----|-----|-----|----|
> Beijing | 0.1 |  3.2 | 35.7  | 41.8 |  100
> Chengdu | 2.2 | 17.9 | 55.5 | 70.4 | 100
> CityIndia|0.8|9.1|60.4|67.6|100
> Harbin | 2.8 | 19.0 | 75.3 | 82.7 | 100
> Porto | 4.1 |20.7 |  60.3 | 72.8| 100
>
> Overall, it is amply clear from the data, that route recommendation is indeed a difficult problem since even the most likely route for a given input often occurs with a low probability.
>
>
> **Comment 1: I wonder if there is a way to estimate the upper bound (see W1) of the accuracy from the data? Showing this estimation can be helpful to understand the nature of the most likely route prediction problem.**
>
> *Answer:* While it is possible to get an upper bound on the individual metrics such as precision, recall and reachability, it is hard to compute a joint upper bound on all metrics. To elaborate, a route that covers all edges in the road network would have a recall of 1. However, such a route is not useful. In general, whether there exists a route that jointly optimizes all three metrics is also unknown.  We also note that the number of possible routes is exponential with the node set size of the road network. Hence, computing the upper bound itself may be NP-hard. At this point, we do not have a conclusive answer to this question. However, we agree that this is an intriguing question and requires further thought.
>
>
> **Comment 2: At line 254, training all the models for up to 36 hours does not seem to be a fair comparison. It would be better to fix the training epochs rather than training time.**
>
> We trained all models till convergence of validation loss. Since all models converged within 36 hours, we wrote this statement. We agree our statement should have been more precise, which we will correct.
>
> To support our statement with empirical data, we show the validation loss of CSSRNN and DeepST against the number of epochs. "-" indicates that the program was terminated due to convergence. As visible, all models converged in terms of validation loss. We will also add this data to the supplementary.
>
> ### CSSRNN
>
> |Epochs | 20 | 40 | 60 | 80 | 100 | 120 | 140 | 160 | 180 | 200 | 220 | 240 | 260 | 280 |
> |----------|-----|------|----|------|-------|------|-------|-------|------|------|--------|-------|-------|------|
> Beijing|10.18  | 6.20 | 7.20| 8.38| 9.61|-|-|-|-|-|-|-|-|-|
> Chengdu | 6.44  |  2.56 |  2.51  |  2.49  | 2.48  | 2.49  | 2.50|-|-|-|-|-|-|-|
> CityIndia| 16.32 | 14.66 | 14.04 | 13.65 | 13.41 | 13.22 | 13.12 | 13.05 | 13.04| 13.07 | 13.16| 13.23 | 13.39 | 13.54 |
> Porto | 8.65| 5.77| 5.46| 5.39 |5.43|5.50|5.57|5.70|5.81|5.91|-|-|-|-|
>
>
> ### DeepST
>
> |Epochs | 20 | 40 | 60 | 80 | 100 | 120 | 140 | 160 | 180 | 200 | 220 | 240 | 260 | 280 |
> |----------|-----|------|----|------|-------|------|-------|-------|------|------|--------|-------|-------|------|
> Beijing| 61.75 |43.18 |35.41 | 32.00|31.11|30.32| 29.93| 30.12|30.63|31.78|-|-|-|-|
> Chengdu| 23.41 |21.25 |20.39 |20.20|20.13|20.23|20.34|20.59||-|-|-|-|-|
> CityIndia|72.07  | 36.86 | 26.87| 23.79| 21.41 |20.21 |19.57| 19.27| 19.10 |19.11 |19.38|20.06|-|-|-|
> Porto | 19.52 |15.37 |13.51 | 12.82 | 12.56| 12.33| 12.49|12.88|-|-|-|-|-|-|
>
>
>
> **Comment 3: How does the proposed approach handle changes in the road network? For example, there could be temporary road closure caused by construction or crashes.**
>
> *Answer:* Since the proposed model is inductive, it can handle changes in the network. Below we present results for this scenario. Specifically, we train the model on road network X, then we randomly add or remove some edges in the road network X to create a new network X'. We then use the model trained on network X to predict on network X'. Addition or removal of edges are done uniformly at random as long as the network remains connected following the deletion. The proportion of additions and deletions is kept equal, i.e., 50% each.
>
> Below are the results on Beijing and Chengdu by NeuroMLR-Greedy. The column headings represent the percentage of edges that were altered. 0% denotes the accuracy in the original network.
>
> ### Chengdu
>
> |Algorithm| 0 | 0.2 | 1.0 | 2.0 |
> |----|-----|-----|-----|-----|
> Precision|86.01 | 85.47 | 84.77 | 84.38 |
> Recall| 83.64 | 83.14 | 80.85 | 79.80 |
> Reachability | 99.94 | 99.45 | 94.13 | 92.24 |
>
>
> ### Beijing
>
>
> |Algorithm| 0 | 0.2 | 1.0 | 2.0 |
> |----|-----|-----|-----|-----|
> Precision | 75.4 | 75.3 | 74.7 | 74.0 |
> Recall | 74.3 | 73.5 | 70.6 | 69.0 |
> Reachability| 99.0 | 96.9 | 90.2 | 86.6 |
>
> Our key insights are as follows:
>
> * **Evaluation:** Evaluation in terms of Precision and Recall is difficult in this scenario as ground truth trajectories do not exist for the new network. To elaborate, all ground-truth trajectories are obtained on the original network. Hence, they would never travel over a new road or take a detour that is required only due to an edge removal. Thus, evaluation metrics are expected to fall even for recommendations that are correct in light of the edge removal/addition.
>
> * **Results:** There is a small drop in accuracy, which is expected as outlined in the discussion above. Nonetheless, the deterioration is small.
>
>
> **Can we adapt to network changes better?** Following road closures, the transition probability distributions may change due to change in mobility patterns. To maintain accuracy levels, the model should adapt to the distribution shift. While one option is to retrain from scratch, it is not practical since (1) it is time consuming and (2) the distribution shift is likely to be minor. For these scenarios, meta-learning allows a potential solution. We hope to pursue this direction in our future work.
>
>
>
>
> **Comment 4: At line 104, should it be Prob. 1 rather than Prob. 2?**
>
> *Answer:* Thank you for pointing this out. Yes, this is an error on our part. We will correct this.

---

> > ### Comment · Reviewer_egXq · 2021-08-31
> > **Thanks for the response**
> >
> > Thanks for the thorough response --- they resolved all my concerns. This is a great work. I would like to raise my rating.

---

### Decision · Program_Chairs · 2021-09-27

**Decision:**

Accept (Poster)

**Comment:**

This paper proposes a method for route recommendation using graph neural networks and Lipschitz embedding. The idea of the proposed method for route recommendation is interesting. The proposed architecture is reasonable. The paper is well written, and experiments are well-designed. As commented by the reviewers, the novelty needs to be clearly explained in the revised paper. The experimental results showed in the author response would strengthen the paper.